# Highly contiguous assemblies of 101 drosophilid genomes

Bernard Y Kim[1†]*, Jeremy R Wang[2†], Danny E Miller[3], Olga Barmina[4], Emily Delaney[4], Ammon Thompson[4], Aaron A Comeault[5], David Peede[6], Emmanuel RR D'Agostino[6], Julianne Pelaez[7], Jessica M Aguilar[7], Diler Haji[7], Teruyuki Matsunaga[7], Ellie E Armstrong[1], Molly Zych[8], Yoshitaka Ogawa[9], Marina Stamenković-Radak[10], Mihailo Jelić[10], Marija Savić Veselinović[10], Marija Tanasković[11], Pavle Erić[11], Jian-Jun Gao[12], Takehiro K Katoh[12], Masanori J Toda[13], Hideaki Watabe[14], Masayoshi Watada[15], Jeremy S Davis[16], Leonie C Moyle[17], Giulia Manoli[18], Enrico Bertolini[18], Vladimír Košťál[19], R Scott Hawley[20], Aya Takahashi[9], Corbin D Jones[6], Donald K Price[21], Noah Whiteman[7], Artyom Kopp[4], Daniel R Matute[6†]*, Dmitri A Petrov[1†]*

[1]Department of Biology, Stanford University, Stanford, United States; [2]Department of Genetics, University of North Carolina, Chapel Hill, United States; [3]Department of Pediatrics, Division of Genetic Medicine, University of Washington and Seattle Children's Hospital, Seattle, United States; [4]Department of Evolution and Ecology, University of California Davis, Davis, United States; [5]School of Natural Sciences, Bangor University, Bangor, United Kingdom; [6]Biology Department, University of North Carolina, Chapel Hill, United States; [7]Department of Integrative Biology, University of California, Berkeley, Berkeley, United States; [8]Molecular and Cellular Biology Program, University of Washington, Seattle, United States; [9]Department of Biological Sciences, Tokyo Metropolitan University, Hachioji, Japan; [10]Faculty of Biology, University of Belgrade, Belgrade, Serbia; [11]University of Belgrade, Institute for Biological Research "Siniša Stanković", National Institute of Republic of Serbia, Belgrade, Serbia; [12]School of Ecology and Environmental Science, Yunnan University, Kunming, China; [13]Hokkaido University Museum, Hokkaido University, Sapporo, Japan; [14]Biological Laboratory, Sapporo College, Hokkaido University of Education, Sapporo, Japan; [15]Graduate School of Science and Engineering, Ehime University, Matsuyama, Japan; [16]Department of Biology, University of Kentucky, Lexington, United States; [17]Department of Biology, Indiana University, Bloomington, United States; [18]Neurobiology and Genetics, Theodor Boveri Institute, Biocentre, University of Würzburg, Würzburg, Germany; [19]Institute of Entomology, Biology Centre, Academy of Sciences of the Czech Republic, Prague, Czech Republic; [20]Department of Molecular and Integrative Physiology, University of Kansas Medical Center, Stowers Institute for Medical Research, Kansas City, United States; [21]School of Life Science, University of Nevada, Las Vegas, United States

*For correspondence:
bernardkim@stanford.edu (BYK);
dmatute@email.unc.edu (DRM);
dpetrov@stanford.edu (DAP)

†These authors contributed equally to this work

Competing interests: The authors declare that no competing interests exist.

**Abstract** Over 100 years of studies in *Drosophila melanogaster* and related species in the genus *Drosophila* have facilitated key discoveries in genetics, genomics, and evolution. While high-quality genome assemblies exist for several species in this group, they only encompass a small fraction of the genus. Recent advances in long-read sequencing allow high-quality genome assemblies for tens or even hundreds of species to be efficiently generated. Here, we utilize Oxford Nanopore sequencing to build an open community resource of genome assemblies for 101 lines of 93

drosophilid species encompassing 14 species groups and 35 sub-groups. The genomes are highly contiguous and complete, with an average contig N50 of 10.5 Mb and greater than 97% BUSCO completeness in 97/101 assemblies. We show that Nanopore-based assemblies are highly accurate in coding regions, particularly with respect to coding insertions and deletions. These assemblies, along with a detailed laboratory protocol and assembly pipelines, are released as a public resource and will serve as a starting point for addressing broad questions of genetics, ecology, and evolution at the scale of hundreds of species.

## Introduction

The rise of long-read sequencing alongside the continuously decreasing costs of next-generation sequencing have served to greatly democratize the process of genome assembly, making it feasible to assemble high-quality genomes at a previously unthinkable scale. Currently, a number of large consortia are leading well-publicized efforts to assemble the genomes of many taxa throughout the Tree of Life. Some often overlapping examples include the Vertebrate Genomes Project (*Rhie et al., 2021*), the Bird 10,000 Genomes Project (*Feng et al., 2020*), the Zoonomia Project (*Zoonomia Consortium et al., 2020*), the Darwin Tree of Life (*Threlfall and Blaxter, 2021*), the Earth Biogenome Project (*Lewin et al., 2018*), and the 5000 Arthropod Genomes Initiative (*Robinson et al., 2011a*). In addition to establishing new standards for modern large-scale genomics projects and opening avenues for genomic research that were previously only feasible in model organisms across a multitude of species, these projects are creating an opportunity to study genetic variation and address fundamental biological questions at a scope that was simply not possible before.

In many respects, the foundation for modern genomics was built by those studying the vinegar (also called fruit or pomace) fly *Drosophila melanogaster* and related species in the family Drosophilidae. As a premier model organism for genetic and biological research since the foundational work of Morgan and colleagues, *D. melanogaster* was, after *C. elegans*, the second metazoan organism to undergo whole-genome sequencing (*Adams et al., 2000*). At that time, the completion of the *D. melanogaster* genome proved the viability of shotgun sequencing approaches and paved the way for larger, more complicated genomes (*Hales et al., 2015*). The genomic tractability that made drosophilids attractive for this work has led to their continued widespread use as model organisms in the genomic era: the whole-genome sequencing of 12 *Drosophila* species (*Clark et al., 2007*) and the characterization of functional elements in *Drosophila* genomes (*Roy et al., 2010*) are prominent milestones in the history of modern genomics.

As it is a popular model system, an extensive collection of genomic resources exists for drosophilids today. Excluding genomes from this study, there are representative genome assemblies available on NCBI databases (GenBank and RefSeq) for about 75 different drosophilid species (*Hotaling et al., 2021*). About a third of these genomes are provided as chromosome-level scaffolds. Along with this diverse catalog of whole-genome sequences are collections of expression and regulation data (*Chen et al., 2014*; *Roy et al., 2010*), maps of constrained (i.e. functional) sequences inferred with comparative genomics tools (*Stark et al., 2007*), and population genomic data (e.g. *Guirao-Rico and González, 2019*; *Lack et al., 2016*; *Signor et al., 2018*). Well-studied *D. melanogaster* was among the first species to have high-quality genomes assembled for multiple individuals, revealing population variation in structural variants (*Chakraborty et al., 2019*; *Long et al., 2018*). Yet even with the intense scientific interest and effort thus far, only a small portion of the remarkably diverse drosophilids, a family which includes over 1600 described and possibly thousands of other undescribed species (*O'Grady and DeSalle, 2018*), is available for genomic study today.

There is much scientific potential to be unlocked by improving the catalog of genomic diversity within this group, and the simplification that long reads bring to the genome assembly process is key. Long reads have proved to be a way to quickly generate affordable yet high-quality genomes, in fact the cost of a highly contiguous and complete *Drosophila* assembly based on long-read sequencing was recently estimated to be about $1,000 US dollars (*Miller et al., 2018*; *Solares et al., 2018*), orders of magnitude less than the first *D. melanogaster* genome. While a number of studies have already used long reads to assemble the genomes of one or a few drosophilid species (*Bracewell et al., 2019*; *Chakraborty et al., 2021*; *Comeault et al., 2020*; *Flynn et al.,*

*2020*; *Hill et al., 2020*; *Mai et al., 2020*; *Miller et al., 2018*; *Paris et al., 2020*; *Rezvykh et al., 2021*; *Solares et al., 2018*), a sequencing and genome assembly project at a scale similar to that of the large genome assembly consortia, especially without similar resources and funding, remains challenging even with the benefits of long reads. Yet, there continue to be rapid improvements to long-read sequencing that may alleviate some of these logistical challenges. Long-read sequencing costs have dropped significantly in the past few years as protocols, kits, and the underlying technology improves. Ultra-long (50–100 kb or longer) reads are obtainable with Oxford Nanopore (ONT) sequencing and under the right conditions should allow entire chromosomes to be fully assembled without additional time-consuming and costly scaffolding methods (e.g. *Nurk et al., 2021*). By simplifying the genome assembly process and reducing the cost of genome assembly even further, these techniques finally make it possible to assemble tens or hundreds of drosophilid genomes at a time.

Here, we present another step toward a comprehensive drosophilid genome dataset: a community resource of 101 de novo genome assemblies from 93 drosophilid species. These genomes were assembled using lines contributed by *Drosophila* researchers from across the world, and represent a diversity of ecologies and geographical distributions. We improve upon the Nanopore-based hybrid assembly (Nanopore plus Illumina) approach for *Drosophila* lines (*Miller et al., 2018*) to substantially increase the sequencing throughput contained in ultra-long reads while reducing overall costs. The contiguity, completeness, and quality of these genomes is assessed. We show that under ideal conditions, about two *Drosophila* lines (assuming an average 180 Mb genome) can be sequenced to at least 30× depth of coverage per ONT r9.4.1 (rev D) flow cell, at an approximate cost of 350 US dollars per line. Along with this manuscript and data, we provide a detailed Nanopore sequencing laboratory protocol specifically optimized for *Drosophila* lines, along with containerized computational pipelines. These genome assemblies and technical resources should facilitate the process of conducting large-scale genome projects in this key model clade and beyond.

## Results and discussion

### Taxon sampling

Our selection of species and strains for sequencing (*Table 1*) improves the geographic, ecological, and phylogenetic diversity of genomic data from the family Drosophilidae. Most (99 of 101) of the genome assemblies presented here are from 14 species groups in subgenera *Drosophila* and *Sophophora* of the subfamily Drosophilinae (*Toda, 2020*). One species of each of the genera *Leucophenga* and *Chymomyza*, both contained in less-studied sister subfamily Steganinae, have also been sequenced. We note some taxonomic inconsistencies arising from the paraphyly or polyphyly of certain drosophilid taxa (*Finet et al., 2021*; *O'Grady and DeSalle, 2018*; *Yassin, 2013*) but will make no attempt to address those issues here. The sequenced species originate from mainland and island locations in North America, Europe, Africa, and Asia; are distributed from northern (e.g. *D. tristis*, *D. littoralis*) to equatorial (e.g. *D. bocqueti*) latitudes; represent two independent transitions to leaf-mining herbivory (*Scaptomyza* and *Lordiphosa*); and for some species, like the pest *Zaprionus indianus*, represent reproductively isolated populations taken from throughout the range. For difficult to culture species, for instance *Leucophenga varia* and some *Lordiphosa* spp., only wild-caught flies were sequenced. Finally, we have sequenced lines in active research use. Additional genomic resources like gene expression or population data should be expected in the near future to accompany many of these assemblies. For species where multiple lines were assembled, we have selected a recommended line to use based on genome quality and denote this recommendation in *Table 1*.

### Near chromosome-scale assembly with ultra-long reads

We sequenced the fly samples using a ONT 1D ligation kit approach, replacing magnetic bead cleanups with size selective precipitation. This modified workflow is optimized for genomic DNA extractions from 15 to 30 whole flies, increases the yield of ultra-long reads relative to the standard ligation kit protocol, increases overall sequencing throughput, and significantly reduces the cost of library preparation. Sequencing runs varied with sample quality and type, and in general read lengths and throughput increased over the course of this work with improved iterations of the protocol. Under optimal conditions and with enough starting material (at least 2,000 ng of very high

**Table 1.** Species and strain information for all samples assembled for this work.

Note: Species group and subgroup information is taken from the NCBI Taxonomy Browser with slight modifications following *O'Grady and DeSalle, 2018*. Strain names along with corresponding NDSSC and Kyoto DGRC stock center numbers are provided to the best of our knowledge. See *Supplementary file 1* and *Supplementary file 6* for detailed information on samples and data. When multiple lines of a species are listed, * denotes the preferred assembly.

| Subgenus | Group | Subgroup | Species | Sex | Strain name | NDSSC | Kyoto DGRC/ Ehime | Additional notes |
|---|---|---|---|---|---|---|---|---|
| *Sophophora* | *melanogaster* | *melanogaster* | *D. melanogaster* | MF | ISO-1 GENOME | 14021-0231.36 | NA | BDGP reference strain |
| | | | *D. mauritiana* | F | NA | 14021-0241.01 | NA | *Miller et al., 2018* |
| | | | *D. simulans* | F | NA | 14021-0251.006 | NA | *Miller et al., 2018* |
| | | | *D. sechellia* | F | NA | 14021-0248.01 | NA | *Miller et al., 2018* |
| | | | *D. teissieri* * | M | 273.3 | NA | NA | |
| | | | *D. teissieri* | M | CT02 | NA | NA | |
| | | | *D. yakuba* | F | NA | 14021-0261.01 | NA | *Miller et al., 2018* |
| | | | *D. erecta* | F | NA | 14021-0224.01 | NA | *Miller et al., 2018* |
| | | *eugracilis* | *D. eugracilis* | F | NA | 14026-0451.02 | NA | *Miller et al., 2018* |
| | | *suzukii* | *D. subpulchrella* | M | L1 | NA | NA | |
| | | | *D. biarmipes* | MF | 361.0 iso1 l-11 GENOME strain 1 | 14023-0361.10 | NA | modENCODE strain |
| | | *takahashii* | *D. takahashii* | F | IR98-3 E-12201 | NA | E-912201 | inbred derivative of Ehime stock IR98-3 |
| | | *ficusphila* | *D. ficusphila* | F | 631.0-iso1 l-10 GENOME | 14025-0441.05 | NA | modENCODE strain |
| | | *rhopaloa* | *D. carrolli* | MF | KB866 | NA | NA | |
| | | | *D. rhopaloa* | MF | BaVi067 GENOME | 14029-0021.01 | E-24701 | modENCODE strain |
| | | | *D. kurseongensis* | F | SaPa58 | NA | NA | |
| | | | *D. fuyamai* | F | KB-1217 | 14029-0011.01 | NA | |
| | | *elegans* | *D. elegans* | F | HK0461.03 GENOME | 14027-0461.03 | NA | modENCODE strain |
| | | *suzukii* | *D. oshimai* | M | MT-04 | NA | NA | |
| | | *montium* | *D. bocqueti* | M | YAK3_mont-66 | NA | NA | |
| | | | *D. sp aff chauvacae* | M | mont_up-71 | NA | NA | |
| | | | *D. jambulina* | MF | st-2 | 14028-0671.01 | NA | |
| | | | *D. kikkawai* | F | 561.0-iso4 l-10 GENOME | 14028-0561.14 | NA | modENCODE strain |
| | | | *D. rufa* | F | EH091 iso-C L_3 | NA | 914802 | inbred derivative of Ehime stock EH091 |
| | | | *D. triauraria* | F | NA | 14028-0691.9 | NA | *Miller et al., 2018*; previously mis-identified as D. kikkawai |
| | | *ananassae* | *D. malerkotliana pallens* | F | palQ-isoG | NA | NA | |
| | | | *D. malerkotliana malerkotliana* | MF | mal0-isoC | 14024-0391.00 | NA | inbred derivative of strain 14024- |

Table 1 continued

| Subgenus | Group | Subgroup | Species | Sex | Strain name | NDSSC | Kyoto DGRC/ Ehime | Additional notes |
|---|---|---|---|---|---|---|---|---|
| | | | | | | | | 0391.00 |
| | | | D. bipectinata | MF | 4-4-2-3-1-1-1-1 BackUp | 14024-0381.04 | NA | Inbred derivative of NDSSC strain |
| | | | D. parabipectinata | MF | par2-isoB | 14024-0401.02 | NA | inbred derivative of strain 14024-0401.02 (now extinct) |
| | | | D. pseudoananassae pseudoananassae | F | Wau 125 | NA | NA | |
| | | | D. pseudoananassae nigrens | F | VT04-31 | NA | NA | |
| | | | D. ananassae | F | 14024-0371.13 | NA | NA | *Miller et al., 2018* |
| | | | D. varians | MF | CKM15-L1 | NA | NA | |
| | | | D. ercepeace | MF | 164-14 | 14024-0432.00 | NA | |
| | obscura | obscura | D. ambigua | M | R42 | NA | NA | isofemale strain from the wild |
| | | | D. tristis | M | D2 | NA | NA | isofemale strain from the wild |
| | | | D. obscura | M | BZ-5 | NA | NA | isofemale strain from the wild |
| | | | D. subobscura | M | Küsnacht | NA | NA | standard laboratory strain |
| | | pseudoobscura | D. persimilis | F | NA | 14011-0111.01 | NA | *Miller et al., 2018* |
| | | | D. pseudoobscura | F | NA | 14011-0121.94 | NA | *Miller et al., 2018* |
| | willistoni | willistoni | D. willistoni (Uruguay) * | M | L-G3 | 14030-0811.17 | NA | |
| | | | D. willistoni | F | NA | 14030-0811.00 | NA | *Miller et al., 2018* |
| | | | D. paulistorum L06 * | M | (Heed) H66.1C | 14030-0771.06 | NA | |
| | | | D. paulistorum L12 | M | L12 | 14030-0771.12 | NA | |
| | | | D. tropicalis | M | (Heed) H65.2 | 14030-0801.00 | NA | |
| | | | D. insularis | M | jp01i | NA | NA | isofemale line from J. Powell |
| | | bocainensis | D. sucinea | M | 49.15 | 14030-0791.01 | NA | |
| | | | D. sucinea** | M | H176.10 | 14030-0761.01 | NA | NDSSC strain is misidentified as D. nebulosa |
| | saltans | saltans | D. saltans | M | (Heed) H180.40 | 14045-0911.00 | NA | |
| | | | D. prosaltans | M | (Heed) H29.6 | 14045-0901.02 | NA | |
| | | neocordata | D. neocordata | M | 2536.7 | 14041-0831.00 | NA | |
| | | sturtevanti | D. sturtevanti | F | H191.23 | 14043-0871.01 | NA | |
| | Lordiphosa | miki | L. clarofinis | MF | Guizhou062018LC | NA | NA | Line inbred for 2 generations |

Table 1 continued

| Subgenus | Group | Subgroup | Species | Sex | Strain name | NDSSC | Kyoto DGRC/ Ehime | Additional notes |
|---|---|---|---|---|---|---|---|---|
| | | | | | | | | in the lab before sequencing |
| | | | L. stackelbergi | MF | UCILTSSapporo052019LS | NA | NA | Pool of 50 wild-caught flies |
| | | | L. magnipectinata | MF | UCKTSapporo052019LM | NA | NA | Pool of 50 wild-caught flies |
| | | fenestrarum | L. collinella | MF | UCKTSapporo052019LC | NA | NA | Pool of 30 wild-caught flies |
| | | | L. mommai | MF | MMSapporo052014LM | NA | NA | |
| Drosophila | Zaprionus | vittiger | Z. nigranus | M | st01n | NA | NA | line derived from wild collection |
| | | | Z. camerounensis | M | jd01cam | NA | NA | isofemale line from J. David |
| | | | Z. lachaisei | M | jd01l | NA | NA | line derived from wild collection |
| | | | Z. vittiger | M | jd01v | NA | NA | isofemale line from J. David |
| | | | Z. davidi | M | jd01d | NA | NA | isofemale line from J. David |
| | | | Z. taronus | M | st01t | NA | NA | line derived from wild collection |
| | | | Z. capensis | M | jd01cap | NA | NA | isofemale line from J. David |
| | | | Z. gabonicus | M | jd01gab | NA | NA | isofemale line from J. David |
| | | | Z. indianus RCR04 | M | RCR04 | NA | NA | |
| | | | Z. indianus 16GNV01 | M | 16GNV01 | NA | NA | |
| | | | Z. indianus BS02 * | M | BS02 | NA | NA | |
| | | | Z. indianus CDD18 | M | CDD18 | NA | NA | |
| | | | Z. africanus | M | BS06 | NA | NA | |
| | | | Z ornatus | M | jd01o | NA | NA | isofemale line from J. David |
| | | tuberculatus | Z. tsacasi | M | car7-4 | NA | NA | |
| | | | Z. tsacasi * | M | jd01t | NA | NA | isofemale line from J. David |
| | | inermis | Z. kolodkinae | M | jd01k | NA | NA | isofemale line from J. David |
| | | | Z. inermis | M | 18BSZ10 | NA | NA | |
| | | | Z. ghesquierei | M | jd01ghe | NA | NA | isofemale line from J. David |
| | cardini | dunni | D. dunni | M | H254.21 | 15182-2291.00 | NA | |
| | | | D. arawakana | M | MONHI050227(B)-104 | 15182-2261.03 | NA | |
| | | cardini | D. cardini | M | NA | 15181-2181.03 | 917701 | |
| | funebris | funebris? | undescribed (Sao Tome mushroom) | M | st01m | NA | NA | undescribed species collected on mushroom, Sao Tome |
| | | funebris | D. funebris | M | fst01 | NA | NA | line derived from wild collection |

Table 1 continued

| Subgenus | Group | Subgroup | Species | Sex | Strain name | NDSSC | Kyoto DGRC/ Ehime | Additional notes |
|---|---|---|---|---|---|---|---|---|
| | immigrans | immigrans | D. immigrans * | F | FK05-19 | 15111.1731.12 | NA | |
| | | | D. immigrans kari17 | M | kari17 | NA | NA | |
| | | (incertae sedis) | D. pruinosa | M | iso-A1 l-9 | NA | NA | |
| | | quadrilineata | D. quadrilineata | M | quad-TMU | NA | 914402 | |
| | tumiditarsus | | D. repletoides | M | ISZ-isoB I-10 | NA | NA | |
| | Scaptomyza | Scaptomyza | S. montana | MF | iso-CA-L1 | NA | NA | |
| | | | S. graminum | F | TMU-2019 | NA | NA | 30 wild-caught females |
| | | Parascaptomyza | S. pallida | MF | iso-CA-L1 | NA | NA | |
| | | Hemiscaptomyza | S. hsui | MF | iso-CA-L1 | NA | NA | |
| | HawaiianDrosophila | orphnopeza | D. sproati | MF | DKPTOMS02 | NA | NA | Pool of wild-caught flies |
| | | | D. murphyi | MF | DKPHETFM01 | NA | NA | Flies from recently established but not inbred lab line |
| | | grimshawi | D. grimshawi | F | NA | 15287-2541.00 | NA | Same line as caf1 genome |
| | virilis | virilis | D. virilis | F | NA | 15010-1051.87 | NA | *Miller et al., 2018* |
| | | | D. americana | M | 3367.1 | 15010-0951.00 | NA | Also called Anderson strain |
| | | | D. littoralis | M | Kilpisjärvi 1 | NA | NA | Originally misidentified as *D. ezoana* (Lankinen 1986, J Comp Physiol A 159: 123-142) |
| | repleta | repleta | D. repleta | M | kari30 | NA | NA | |
| | | mulleri | D. mojavensis | F | 15081-1352.22 | NA | NA | *Miller et al., 2018* |
| genus: Leucophenga | | | L. varia | M | nc01v | NA | NA | Sequenced single wild-caught fly, no amplification |
| genus: Chymomyza | | | C. costata | M | Sapporo | NA | NA | |

* denotes the genome of best quality when multiple assemblies are available for a species.

molecular weight DNA) to prepare at least three library loads (~1200–500 ng total prepared library, 350–500 ng per load), along with regular DNAse flushes to maintain yields, Nanopore sequencing runs following the supplied protocol should net 12–15 Gb of data per R9.4.1 flow cell with a read N50 greater than 20 kb, and about 30% of data in reads longer than 50 kb. We generated paired-end, 150 bp Illumina reads for most strains unless public datasets were available.

Deep (average 52×) sequencing coverage with a substantial fraction of ultra-long reads (*Supplementary file 1*) resulted in high-quality genome assemblies that were comparable to and often better than currently available reference genomes in terms of contiguity and completeness (*Figure 1*, *Figure 1—figure supplement 1*, *Supplementary file 2*). We chose Flye (*Kolmogorov et al., 2019*) as our assembler based on superior contiguity and favorable runtimes relative to Miniasm (*Li, 2016*) and Canu (*Koren et al., 2017*; *Figure 1—figure supplement 2*). To provide standardization for measures of contiguity, we estimated genome size for each assembly using long-read coverage over single-copy BUSCO loci (*Supplementary file 2*).

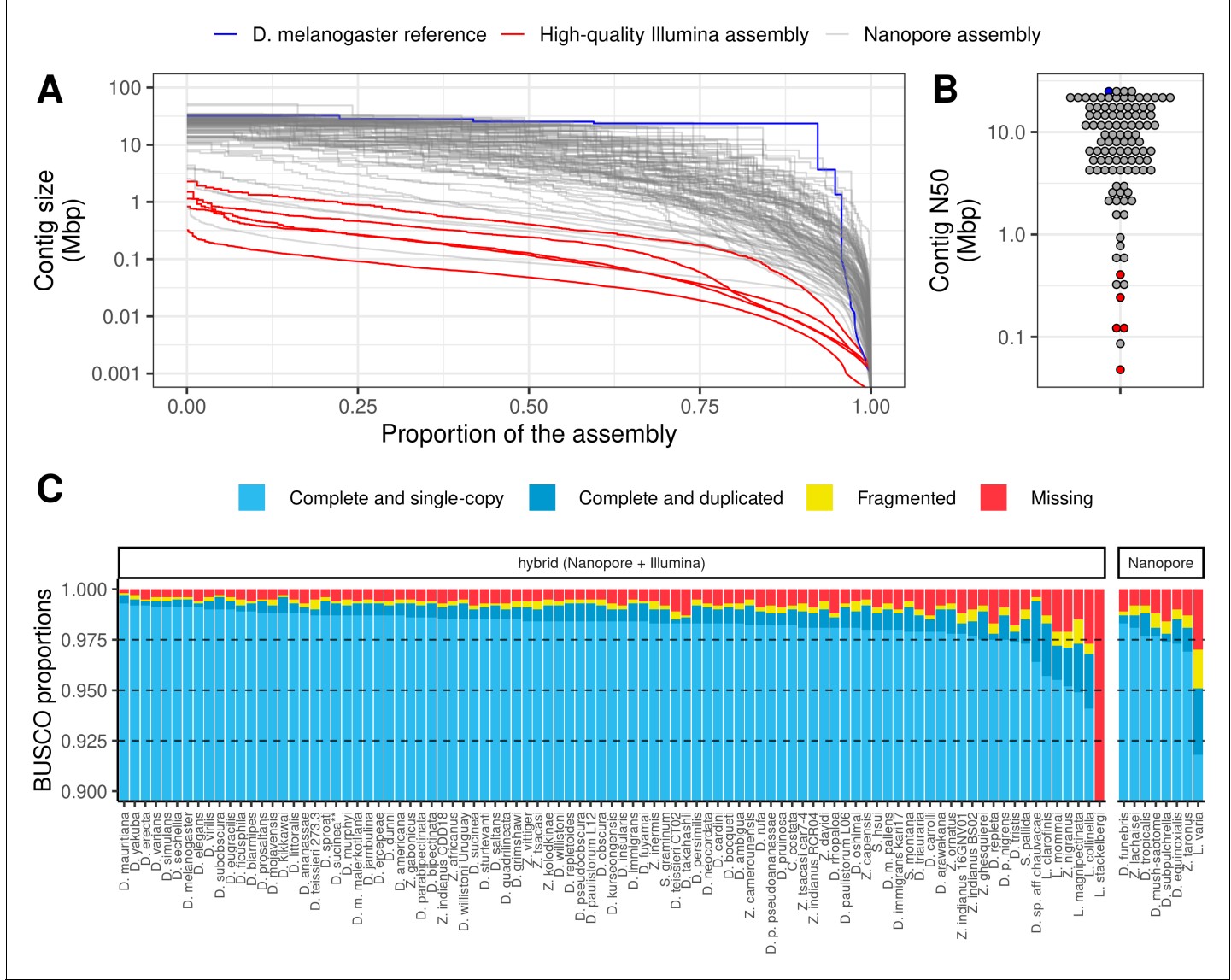

**Figure 1.** Nanopore-based assemblies are highly contiguous and complete. (**A,B**) Assembly contiguity is compared to the *D. melanogaster* v6.22 reference genome (blue) as well as five recently published, highly contiguous Illumina assemblies (red lines, *D. birchii, D. bocki, D. bunnanda, D. kanapiae, D. truncata*; *Bronski et al., 2020*). (**A**) *Nx* curves, or the (y-axis) size of each contig when contigs are sorted in descending size order, in relation to the (x-axis) cumulative proportion of the genome assembly that is covered. (**B**) The distribution of contig N50, the size of the contig at which 50% of the assembly is covered. (**C**) Assembly completeness assessed by BUSCO v4.0.6 (*Seppey et al., 2019*). Note, *D. equinoxialis* was evaluated with BUSCO v4.1.4 due to an issue with v4.0.6. *L. stackelbergi* has >10% missing BUSCOs. Individual assembly summary statistics are provided in *Supplementary file 2*.

The online version of this article includes the following figure supplement(s) for figure 1:

**Figure supplement 1.** Nanopore-based assemblies compare favorably to representative genomes on NCBI.

**Figure supplement 2.** Large improvements in assembly contiguity from an updated assembly workflow.

**Figure supplement 3.** Contiguity metrics standardized by the estimated genome size.

**Figure supplement 4.** Estimated genome size is similar to assembly size.

Of 101 total assemblies, 94 contain over 98% of the assembly in contigs larger than 10 kb, and both contig N50s and NG50s exceed 1 Mb for these genomes (*Figure 1A*, *Figure 1B*, *Figure 1—figure supplement 3*, *Supplementary file 2*). Assembly sizes were highly correlated with estimated genome sizes (*Figure 1—figure supplement 4*). In addition to meeting the megabase contig N50 standard for new genomes proposed by the Vertebrate Genomes Project (*Rhie et al., 2021*), these

statistics show that most of the genome is present in the assembly in megabase-sized contigs. In other words, the assemblies are nearly at the chromosome level. For comparison, of the 76 representative drosophilid genomes that were previously available on NCBI (*Hotaling et al., 2021*), only 25 have an N50 greater than 1 Mb (*Figure 1—figure supplement 1*). Moreover, many of these highly contiguous NCBI genomes are scaffolded, an additional step that would have added a significant amount of time and additional expenses to this study. Even when DNA was extracted from pools of wild-caught flies or a single fly (*Leucophenga varia*) resulting in sub-optimal read lengths and output, the assembly was comparable to existing short read assemblies (*Figure 1A*, *Figure 1B*). High contiguity resulted in benchmarking universal single-copy ortholog (BUSCO) completeness (*Seppey et al., 2019*; *Simão et al., 2015*) in the range of 97–99+% for all but the three most fragmented genomes (*Figure 1C*). As with contiguity, the completeness of these genomes is comparable to reference genomes on NCBI (*Figure 1—figure supplement 1*).

## Estimates of sample diversity

We have utilized a variety of fly samples, from highly inbred lab lines to wild-caught flies, for genome assembly. We therefore sought to quantify the level of diversity inherent to each sample and use variant calls to estimate the error rate for each assembly. Long and short reads (if available) were mapped separately to each finished genome and variant calling was performed with PEPPER-Margin-DeepVariant (*Shafin et al., 2021*) for long reads and BCFtools (*Danecek et al., 2021*; *Li, 2011*) for short reads. After quality filtering and masking genomic regions annotated as repeats, the counts of single nucleotide polymorphisms (SNPs), indels, and the fraction of sites with a non-reference SNP were computed (*Figure 2*, *Supplementary file 3*). Note, when short reads were not from the same strain as used for the assembly, short read polishing was used to only correct indels, and called SNPs will not accurately represent the variation in the sample that was sequenced with Nanopore.

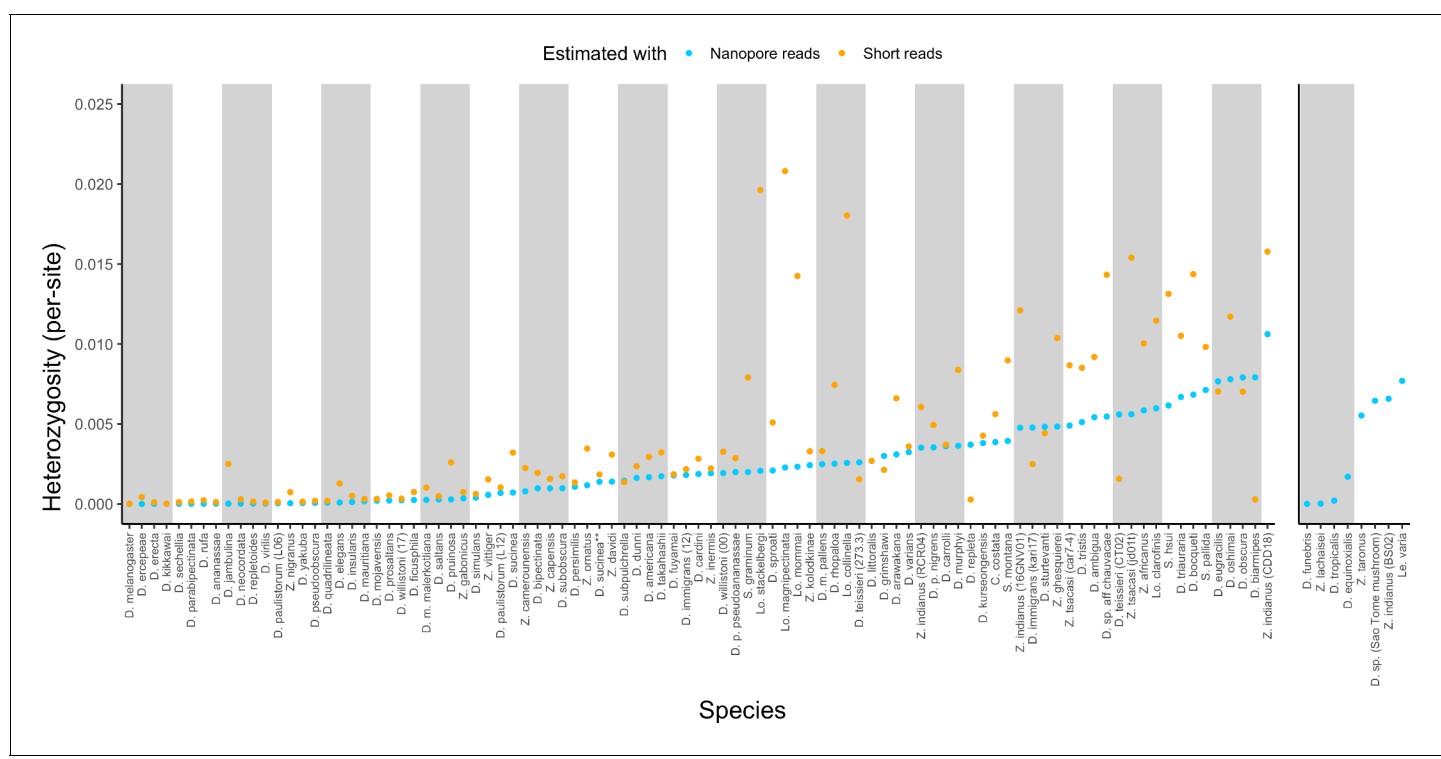

**Figure 2.** Estimated heterozygosity in the data used for genome assembly. Per-site SNP heterozygosity (number of heterozygous SNPs/number of callable sites) is plotted for each of the 101 assembled lines. Blue dots represent heterozygosity estimates from Nanopore reads with PEPPER-Margin-DeepVariant (*Shafin et al., 2021*). Orange dots represent heterozygosity estimates from short reads with BCFtools (*Li, 2011*). The genomes on the right are for species that did not have available short-read data. Numerical values for these estimates are provided in *Supplementary file 4*.
The online version of this article includes the following figure supplement(s) for figure 2:

**Figure supplement 1.** Assembly contiguity is not related to sample heterozygosity.

Also note that SNP calls from Nanopore data should be relatively accurate but indel calls will not (*Shafin et al., 2021*).

Large variation in sample diversity over several orders of magnitude was observed. Estimated SNP heterozygosity, the number of heterozygous SNPs divided by the number of callable sites, ranged from 0.00035% to 1.1% from long reads and 0.0015% to 2.1% from short reads, and heterozygosity estimated from long reads was systematically lower than that from short reads, particularly when sample diversity was high (*Figure 2*, Figure S6). Qualitative patterns of heterozygosity generally tracked the history of the samples (e.g. the highly inbred reference strains had very low diversity). Conditioning on datasets where both long and short reads were generated from the same sample, heterozygosity estimates from both types of reads were positively correlated (Pearson correlation $R^2$=0.50, p=1.13×10–12). If we ignore *Lordiphosa*, the group with wild-caught or recently collected samples that was consequently the most challenging to assemble, this correlation is greatly increased (Pearson correlation $R^2$=0.81, p<2.2×10–16). Interestingly, we did not observe a significant relationship (p=0.30) between estimated heterozygosity and assembly contiguity (*Figure 2— figure supplement 1*). The number of heterozygous non-reference variants almost always exceeded the number of homozygous variants (*Supplementary file 3*), as would be expected from residual diversity in the sequenced lines.

## Estimates of sequence quality

Next, we estimated the genome-wide error rates in our assemblies using both the variant calls obtained previously and a reference-free method (*Supplementary file 4*). For the first approach, Phred-scaled (*Ewing et al., 1998*) consensus quality (QV) was estimated by assuming all sites with a non-reference variant were an error. The error rate was then computed by dividing the number of sites with at least one non-reference variant by the total number of callable bases. As expected from the patterns of heterozygosity estimated from long and short reads, there was a large amount of variability in quality scores. Estimates from short reads ranged from QV17 to QV45 and from long reads were slightly higher, from QV19 to QV52 (*Supplementary file 4*).

This method is likely to be biased by assembly features that affect the quality of read mapping, for example, we remove sequences annotated as repeats when filtering the variant calls. To address this bias, we employed the reference-free approach implemented in Merqury (*Rhie et al., 2020*) for the 94 assemblies which had some kind of short-read data available (*Figure 3A*, *Supplementary file 4*). Estimated quality scores ranged from QV16 to QV40, and once again, samples for which reads from a different strain or a genetically diverse sample (i.e. wild samples or recent isolates) were used had the lowest estimated QV. Merqury-estimated QV was on average higher than consensus quality estimated by the variant calling methods, but the relative ranking of QV estimates remained largely consistent with QV based on short-read (Spearman's $\rho$=0.642, p<2.2e-16) and long-read (Spearman's $\rho$=0.684, p<2.2e-16) variant calls.

While these estimates showed our genomes to mostly fall below the often-recommended QV40 threshold for reference genomes (*Koren et al., 2019*; *Rhie et al., 2021*), there are many reasons to expect that sequence quality in certain regions of the genome will be far better than the average. As expected, we found that QV estimates were particularly low when short-read data from a different sample was used for the estimation, as any true variation between strains will inflate the error rate. Because we sequenced pools of flies, residual polymorphism will be found in the data even when long and short reads are sampled from the same pool of flies. In these cases QV might be considered as a lower bound estimate of the true accuracy of the assembly. Additionally, complex coding sequences are likely to be far more accurate than other regions of the genome, like repeats, due to better short-read mapping. The single genome-wide estimates of QV we report obscure this variation.

## Nanopore-based assemblies are highly accurate in coding regions

For these reasons, we found it critical to further examine how errors are distributed in Nanopore assemblies. Of particular concern is the accuracy of coding sequences. Gene annotation is an important and obvious next step after assembling a new genome, but Nanopore sequences are known to systematically contain indels in homopolymer runs that cannot be called accurately when a run exceeds the size of the nanopore reader head. Indel disruptions to otherwise highly accurate coding

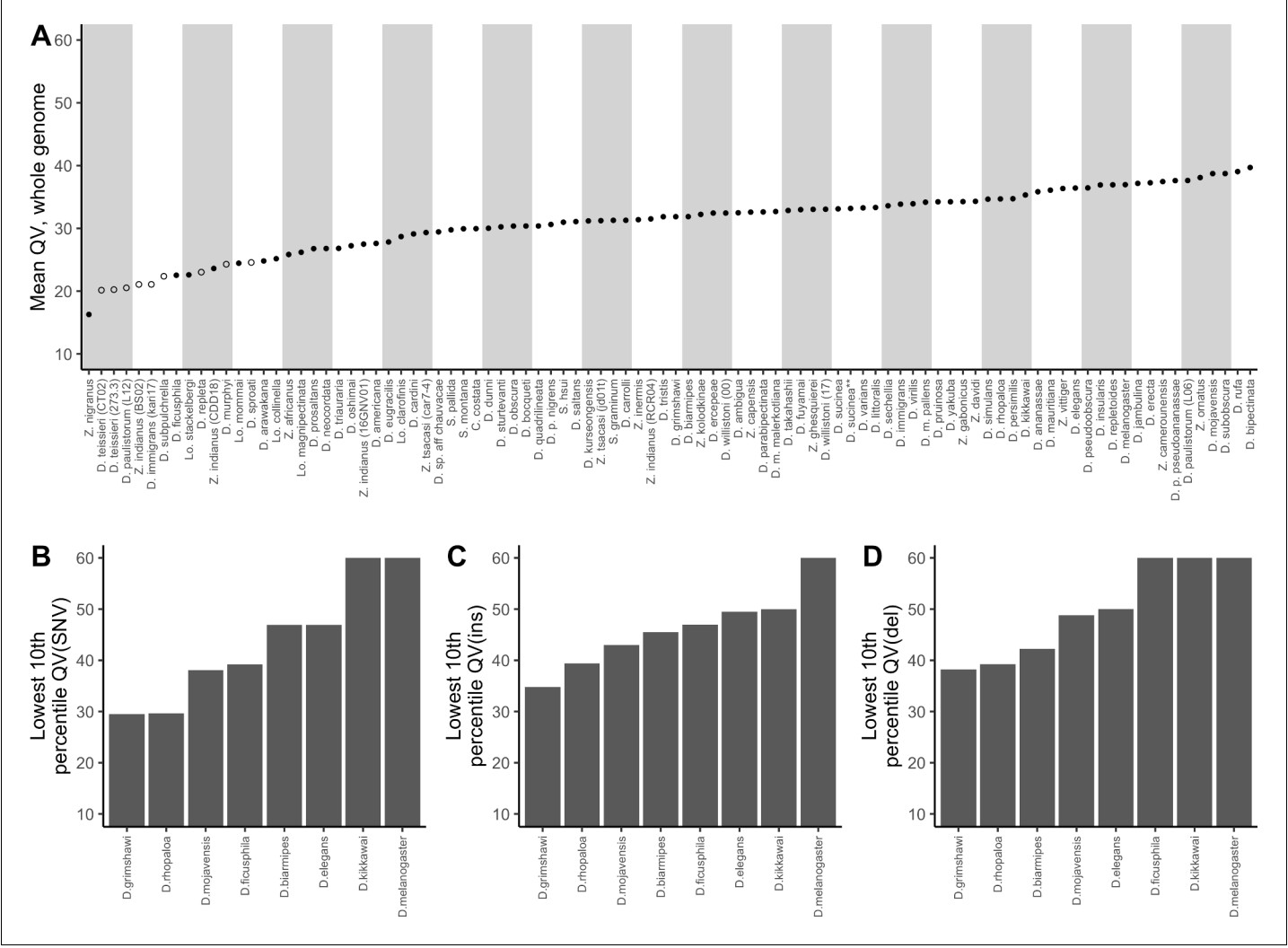

**Figure 3.** Nanopore-based *Drosophila* assemblies are accurate, particularly in coding regions. (**A**) Genome-wide, Phred quality scores estimated with the reference-free, k-mer based approach implemented in Merqury (***Rhie et al., 2020***). Merqury requires a short-read dataset to perform the evaluation. Filled circles represent QV estimates with short-read data from the same strain used for Nanopore sequencing, and empty circles denote estimates using short-read data from a different strain than used for Nanopore sequencing. (**B, C, D**) Phred quality score cutoffs for the bottom 10th percentile of 100 kb genomic windows, as evaluated with a reference-based approach, in coding sequences only. Quality scores are capped at 60 for visualization purposes. At least 90% of 100 kb windows are this accurate. Only Nanopore assemblies with an NCBI RefSeq genome counterpart of the same strain were evaluated. Accuracy is shown for SNVs (**B**), insertions (**C**), and deletions (**D**) separately. Additional details on quality score estimates are provided in ***Figure 3—figure supplement 1*** and ***Supplementary file 4***.

The online version of this article includes the following figure supplement(s) for figure 3:

**Figure supplement 1.** Variation in sequence accuracy within the genome assemblies.

**Figure supplement 2.** Large insertions account for nearly all differences between the Nanopore-based and reference *D. melanogaster* assembly.

sequences would have a disproportionately large negative impact on protein prediction (***Watson and Warr, 2019***). On the other hand, it is likely that coding sequences are generally more accurate than the rest of the genome since short-read mapping is generally more reliable there. In theory, most exons should be free of errors somewhere between a genome-wide quality of QV30 to QV40 (***Koren et al., 2019***), but many of our assemblies do not appear to reach this benchmark.

Reference-based quality assessments were used to better understand how error rates vary across different genomic elements. We downloaded the 8 NCBI RefSeq genome assemblies for which we had a Nanopore genome of the same species and strain: *D. biarmipes, D. elegans, D. ficusphila, D. grimshawi, D. kikkawai, D. melanogaster, D. mojavensis*, and *D. rhopaloa*. Using the ONT Pomoxis

software, we aligned each Nanopore assembly to its corresponding reference genome and estimated QV in non-overlapping 100 kb windows, using the entire sequence, then only coding sequences, introns, intergenic regions, and repeats, using gene and repeat definitions provided through NCBI RefSeq. All differences between query and reference assemblies were considered to be errors.

As expected, we found that sequence accuracy varied greatly within each genome assembly (*Figure 3—figure supplement 1*). Mean genome-wide QV ranged from QV15 to QV24 while median QV across the 100 kb windows ranged from QV14 to QV36. When looking only at coding sequences, mean QV ranged from QV23 to QV29, while the median window accuracy, with the exceptions of *D. grimshawi* (QV25) and *D. rhopaloa* (QV30), indicated complete identity (>QV50) between assembly and reference. For *D. grimshawi* and *D. rhopaloa*, SNVs were the primary contributor to the error rate and the number of indels was similar to the other genomes (median QV(indel)>50). Sequence accuracy was lower when looking at introns, intergenic regions, and repeats, in that order. However, regardless of the genomic element type, median QV across the windows always exceeded mean QV, often by more than QV10, or an order of magnitude difference in the error rate. In other words, differences between Nanopore and reference assemblies were clustered heavily into a few genomic regions, and most coding sequences were very accurate despite the seemingly high mean error rate (*Figure 3B*, *Figure 3C*, *Figure 3D*). Further caution is warranted in the interpretation of these quality scores: we have assumed that all differences between our Nanopore-based genomes and the reference genomes are errors in the Nanopore assembly, rather than errors in the reference, or true differences between the two sequenced samples. We will shortly show that reference-based comparisons might be heavily biased against the Nanopore assemblies.

To better understand the nature of putative indel errors in coding sequences, we focused on the *D. melanogaster* reference strain, where we have the best information about the genome from multiple independent high-quality assemblies (*Kim et al., 2014*; *Koren et al., 2017*; *Solares et al., 2018*). While *D. melanogaster* is a best-case scenario for genome assembly with a fly line, we think it reasonable to expect errors in other assemblies, for which we have utilized the same genome assembly workflow, to be similar in nature. Across the 22,209,264 bp of our *D. melanogaster* genome that aligned to reference coding sequences, our assembly contained 15 insertions and 17 deletions in 21 out of 13,913 (0.15%) queried protein-coding genes, with a total of 10,092 inserted and 46 deleted base pairs relative to the reference. All deletions (15 out of 15) were under 50 bp, 8 out of 15 insertions were under 50 bp, and the remaining 7 out of 15 insertions ranged from 120 bp to 4410 bp (*Figure 3—figure supplement 2*, *Supplementary file 5*). These larger insertions account for nearly all (99.3%) of the coding sequence differences between our genome and the reference. There is a clear, disproportionate impact of these large insertions in an otherwise nearly identical protein-coding genome.

We followed up on each of these 32 coding indels through manual curation with the genome browser IGV (*Robinson et al., 2011b*). Using the Release 6 *D. melanogaster* genome (*Hoskins et al., 2015*) as the reference, we aligned Nanopore and Illumina reads, a different Nanopore-based de novo assembly of the reference strain (*Solares et al., 2018*), a PacBio-based de novo assembly (*Kim et al., 2014*; *Koren et al., 2017*), and our assembly. We were particularly interested in the two other long-read assemblies, as we wondered if they might independently support any of the large variants in our assembly. RepeatMasker annotations for the Nanopore-based assembly were lifted over into Release 6 coordinates to see if these indels overlapped with a repetitive element.

This manual curation process revealed that the coding indels, in addition to being exceedingly rare, could be straightforwardly explained by regions of poor short read mapping and the presence of a duplicate contig in the assembly (*Supplementary file 5*). A series of large and small indels, including four out of the five insertions longer than 100 bp, overlapped a tandem repeat in genes *CR44666*, *Mu68Ca*, and *Mu68E*. While short reads mapped poorly to this region, limiting our ability to determine accuracy locally, long reads spanning the entire region and the two other long-read assemblies supported the large insertions. The remaining long (1414 bp) insertion was similarly supported by long reads and the other assemblies, but did not overlap with a repeat. Again, these insertions account for more than 99% of the indel differences between our genome and the reference. The remaining indels occurred in either repetitive regions (simple repeats and long interspersed nuclear element retrotransposons), in homopolymer runs in regions with poor short read mapping, or along a single contig that appeared to be a short duplicated segment of chromosome

2L. The other contig was error-free. All indels occurred on contigs with poor short-read mapping, suggesting they were a consequence of locally ineffective short read polishing, but also that sensible filtering based on short read depth or map quality would prevent these issues from propagating into downstream analyses. Importantly, these results suggest that reference-based quality analyses can be heavily biased against long-read assemblies and further support our caution against a naive projection of genome-wide quality score estimates onto coding regions.

## A comparative genomics resource

To demonstrate the potential this dataset holds for the study of genome evolution and chromosome organization, we revisit a classic result with our highly contiguous assemblies. Although the ordering of genes in drosophilid chromosomal (Muller) elements has been extensively shuffled throughout ~53 million years of evolution (*Suvorov et al., 2021*), the gene content of each element remains largely conserved (*Bracewell et al., 2019*; *Ranz et al., 2001*; *Sturtevant and Novitski, 1941*). To examine synteny in our assemblies, many of which contain several contigs tens of megabases in length, we constructed an undirected graph using single-copy orthologous markers (i.e. BUSCOs). The number of times two markers were connected by assemblies determined the weight of the graph's edges. A graph layout method was applied to spatialize (map) these relationships, clustering together BUSCOs that are frequently connected in the assemblies. We found that BUSCOs formed six major clusters following the *D. melanogaster* chromosome arm on which they are found, consistent with the expected conservation of gene content in Muller elements across drosophilids (*Figure 4*). Furthermore, the lack of a clear order within groups is consistent with extensive shuffling within Muller elements. This demonstrates that our dataset can be used for studies of genome evolution. New reference-free, whole-genome alignment methods (*Armstrong et al., 2020*) should substantially facilitate more detailed comparative analyses.

## Repeat content

A large number of genome assemblies enables comparative analysis of repeat variation against a wide range of genome assembly sizes (140–450 Mb), for example the independent expansions of satellite repeats in *D. grimshawi* or retroelements in *D. paulistorum*, *D. bipectinata*, or *D. subpulchrella* (*Figure 5*). Within our dataset alone, RepeatMasker annotations reveal large variation in repeat content among drosophilids (*Figure 5*). No correlation exists between assembly contiguity and repeat content (*Figure 5—figure supplement 1*), suggesting long-read sequencing overcomes many of the challenges to drosophilid genome assembly posed by repetitive sequences. Additionally, we observe a positive relationship between the size of repetitive sequences and non-repetitive sequences, suggesting that genome size is influenced by expansions and contractions of both portions of the genome (*Figure 5—figure supplement 2*). Some discretion is warranted in the interpretation of these results. Repeats are likely to be better annotated in genomes from well-studied species groups, since they are more likely to be well-characterized in the repeat databases we used. Nevertheless, the high continuity of these assemblies should allow for the proper identification of new transposable elements in the genomes and allow for the analyses of transposable element evolution at the level of individual transposable elements or transposable element families in a way that is not feasible with more fragmented genome assemblies (*Clark et al., 2007*).

## Next steps

We have built an open resource of 101 nearly chromosome-level drosophilid genome assemblies, adding to the rapidly growing number of high-quality genomes available for this model system (*Hotaling et al., 2021*; *Suvorov et al., 2021*). We envision this dataset being used to address a large number of outstanding questions entailing large comparative analyses among species, including the comparison of population genomic data between a large number of species, providing unprecedented resolution to investigate fundamental questions about the evolutionary process. In addition, we provide detailed laboratory and computational workflows that we hope will provide a jumping off point for future genome assembly projects in drosophilids or other taxa. While we hope this to already be a valuable resource to the scientific community, we acknowledge there is much to be done to build upon the resource and to improve its usability.

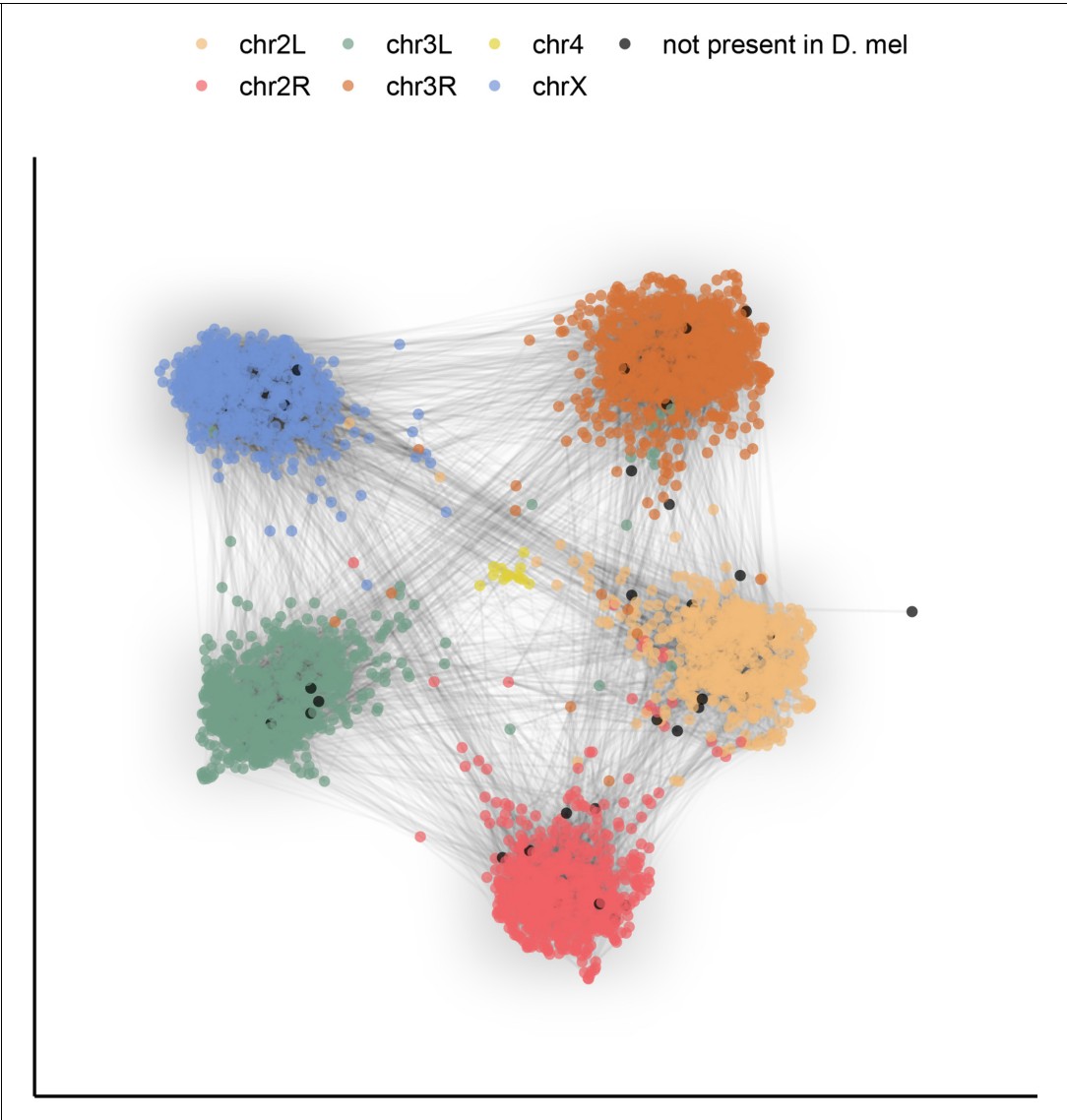

**Figure 4.** Gene content of Muller elements is conserved across drosophilids while gene order changes. Each node in this graph represents an orthologous marker corresponding to single-copy orthologs annotated by BUSCOv4 (*Seppey et al., 2019*). An edge between two nodes represents the number of times that BUSCO pair is directly connected within an assembly. Each BUSCO is colored by the chromosome arm in *D. melanogaster* that it is found on. The ForceAtlas2 (*Jacomy et al., 2014*) graph layout algorithm was used for visualization.

Despite our best efforts to improve species diversity, both the species we sequenced and the drosophilid genomes available today are significantly biased towards well-studied, easy-to-maintain species already in use for scientific research. Reducing sampling bias, with respect to phylogenetic diversity, geographic distribution, and ecology should be a goal of future genome assembly projects in this group. The high input DNA requirement for PCR-free long-read sequencing is a major limitation of our assembly workflow in this context. Our protocol requires a DNA extraction from multiple flies, ideally from an inbred line to minimize genetic diversity in the sample used for assembly. High diversity in the sample usually results in a fragmented assembly with many duplicated sequences, and while these issues can be addressed with computational tools, the quality of the final assembly is still affected. However, some species, for instance the *Lordiphosa* spp. or Hawaiian *Drosophila* we sequenced, cannot be quickly raised in the lab on standard media and thus cannot easily be inbred like other drosophilids. Many other species are simply understudied and sample availability is limited to a few flies collected from the wild and possibly preserved in ethanol for many years. Methods for

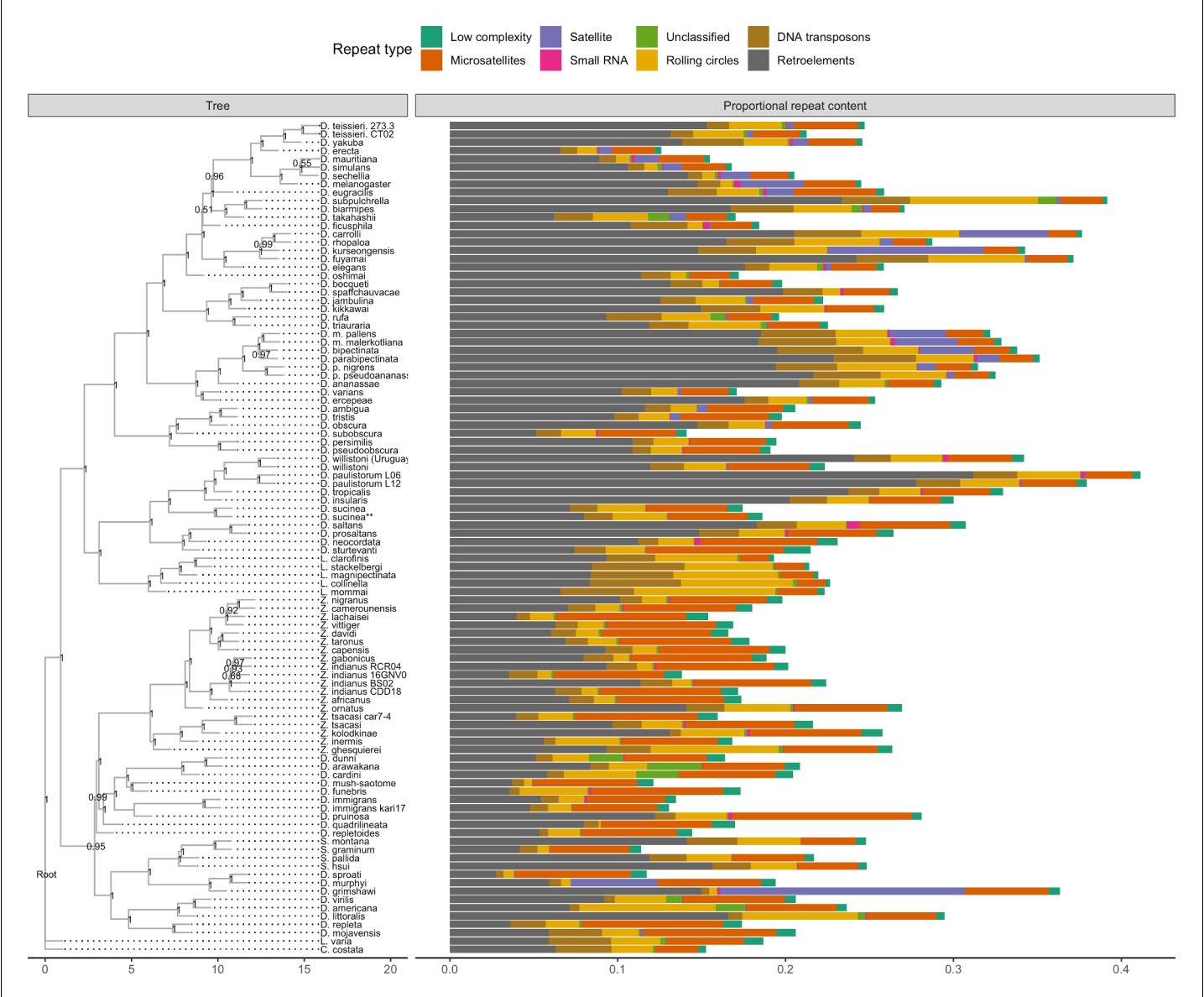

**Figure 5.** Repeat content varies greatly between drosophilid groups. For each species, the proportion of each genome annotated with a particular repeat type is depicted. Species relationships were inferred by randomly selecting 250 of the set of BUSCOs (*Seppey et al., 2019*) that were complete and single-copy in all assemblies. RAxML-NG (*Kozlov et al., 2019*) was used to build gene trees for each BUSCO then ASTRAL-MP (*Yin et al., 2019*) to infer a species tree. Repeat annotation was performed with RepeatMasker (*Smit et al., 2013*) using the Dfam 3.1 (*Hubley et al., 2016*) and RepBase RepeatMasker edition (*Bao et al., 2015*) databases. ASTRAL local posterior probabilities are reported at each node.

The online version of this article includes the following figure supplement(s) for figure 5:

**Figure supplement 1.** Assembly contiguity is not determined by repeat content.

**Figure supplement 2.** The non-repetitive and repetitive portions of the genome both contribute to genome size differences between drosophilids.

assembling genomes with small quantities of DNA from single insects (*Adams et al., 2020*; *Kingan et al., 2019*; *Schneider et al., 2021*) or dealing with degraded specimens from older collections will be particularly important as the scope of future work expands beyond stock center and laboratory lines.

Some of these sequencing challenges will be better addressed by new technology and techniques. While we hesitate to make specific recommendations due to the rapidly changing landscape of long-read sequencing and genome assembly methods, there are a few clear ways in which many recently assembled long-read genomes can be improved. Even in the short time since we performed

the sequencing for this work, there have been remarkable improvements to library preparation workflows, the accuracy of base calling algorithms, and assembly tools. At a minimum, we plan to iteratively update these assemblies using newer base calling methods to maximize the usefulness of the dataset.

This alone is unlikely to future-proof Nanopore R9 flow cell-based assemblies when the ultimate goal is to build genomes that are free of errors, and we recommend that a genome assembly project initiated today look beyond a Nanopore and Illumina approach. There is ample room to reduce the per-assembly cost while improving both contiguity and accuracy. The current major obstacle to high genome-wide accuracy is the difficulty of calling bases accurately in homopolymer runs combined with the limitations of short reads for correcting these errors when they occur in genomic regions with poor short read mappability. One new strategy to address this is to generate supplementary lower coverage data from high fidelity long read sequencing, for instance with PacBio HiFi (Nanopore versions are currently in development). New polishing tools are specifically designed to polish Nanopore assemblies with higher-fidelity reads (e.g. *Shafin et al., 2021*) and users should see greatly improved overall sequence accuracy.

This kind of hybrid long-read assembly approach may prove to be even more efficient than the assembly workflow we have presented. Interestingly, we find that high contiguity can be achieved

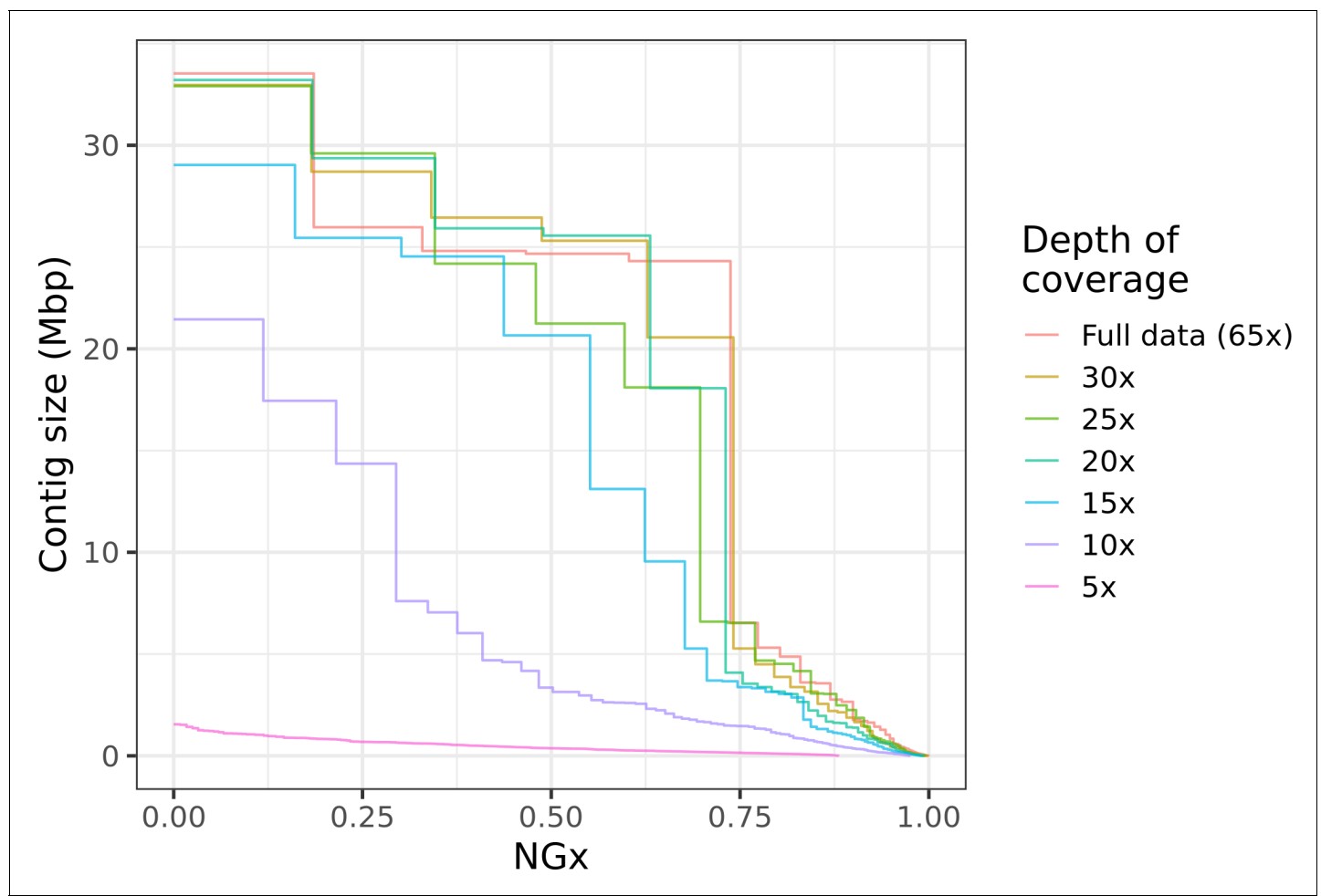

**Figure 6.** Highly contiguous assemblies can be obtained with lower coverage of ultra-long reads. The NGx curve is shown for *Drosophila jambulina* assemblies at varying levels of coverage. The length of the assembly with the full data is assumed to be the genome size. Read sets used for each assembly were obtained by randomly downsampling the basecalled reads (read N50 ~27.5 kb) to varying (5× to 30×) depth of coverage. Proportionally, these read sets contain ~55% of total sequenced bases in reads longer than 25 kb, ~25% of bases in reads longer than 50 kb, and ~7% of bases in reads longer than 100 kb. Near chromosome scale assemblies (N50>20Mb) were achievable even at 15× to 20× depth with this read length distribution. This corresponds to approximately 8× to 10× depth in reads longer than 25 kb.

even with minimal (10–20×) coverage of moderately long (read N50 25 kb) Nanopore reads (*Figure 6*). Similar coverage with even longer reads could serve as a cheap way to generate almost chromosome-level contigs, which will then be polished with higher fidelity long reads or Illumina reads. Ultra-long Nanopore sequencing is also significantly more accessible than before. Recently (as of March 2021), Oxford Nanopore and Circulomics released new ultra-long sequencing kits that, under ideal conditions, allows users to perform ultra-long Nanopore sequencing runs where read N50s exceed 100 kb while nearly doubling overall flow cell throughput compared to the sequencing runs performed for this study. Further cost savings should be possible if sequencing is done with the ONT PromethION or PacBio CLR, depending on the scale of the project. Both technologies have a lower per-base cost than MinION sequencing and similarly long reads can be obtained.

Finally, we are in the process of improving the utility of this resource by generating a suite of comparative genomics tools and annotations to be released in the upcoming months. Specifically, we are utilizing Progressive Cactus (*Armstrong et al., 2020*), a reference-free whole-genome aligner that is designed to be scalable to modern genomic datasets and that has already been applied to hundreds of mammal and bird genomes generated by the Zoonomia (*Zoonomia Consortium et al., 2020*) and Bird 10K (*Feng et al., 2020*) projects. These alignments will be used to create sequence conservation maps (*Hickey et al., 2013*; *Pollard et al., 2010*; *Siepel et al., 2005*), the precision of which should be close to single nucleotide resolution given the large number of drosophilid genomes that are now available. While ultimately RNA-seq across all species will be needed for annotation, we plan to quickly generate the first round of gene annotations using comparative annotation tools. For new assemblies where a previously annotated reference genome is available, LiftOff (*Shumate and Salzberg, 2020*) provides a way to quickly transfer annotations to a new genome. For the more challenging task of gene annotation in species that do not already have a well-annotated reference, we are using the Comparative Annotation Toolkit (*Fiddes et al., 2018*), software to perform first-pass annotations assisted by homology information from the Progressive Cactus alignment. New RNA-seq data will be generated for select species in clades without a well-annotated member (e.g. *Zaprionus*). These tools will provide a framework for anyone to apply iterative improvements as new data become available.

## Reproducibility

Detailed laboratory protocols, computational pipelines, and computational container recipes are provided as a reference and to maximize reproducibility. The protocol is publicly available at Protocols.io and pipeline scripts along with associated compute containers are provided in a public GitHub repository. See Materials and methods for additional details on compute containers, accession numbers, and web links to these resources.

# Materials and methods

**Key resources table**

| Reagent type (species) or resource | Designation | Source or reference | Identifiers | Additional information |
|---|---|---|---|---|
| Strain, strain background (*Drosophila* spp. and relatives) | See *Table 1* and *Supplementary files 1–6* for sample information, strain designations, stock center line identifiers (when applicable), biomaterial provider, and NCBI accession numbers. | | | |
| Commercial assay or kit | Blood and Cell Culture DNA Mini Kit | Qiagen | cat # 13323 | |
| Commercial assay or kit | Ligation Sequencing Kit | Oxford Nanopore | SQK-LSK109 | Superseded by SQK-LSK110 |
| Commercial assay or kit | Flow cell wash kit | Oxford Nanopore | EXP-WSH003 | Superseded by EXP-WSH004 |
| Commercial assay or kit | Short Read Eliminator kit | Circulomics | SKU # SS-100-101-01 | |
| Commercial assay or kit | Companion Module for ONT Ligation Sequencing | NEBNext | cat # E7180S | |

*Continued on next page*

*Continued*

| Reagent type (species) or resource | Designation | Source or reference | Identifiers | Additional information |
|---|---|---|---|---|
| Commercial assay or kit | Nextera XT DNA Library Preparation Kit | Illumina | cat # FC-131–1002 | Superseded by version 2 |
| Commercial assay or kit | Kapa HyperPrep Kit | Roche | cat # KK8502 | |
| Software, algorithm | Flye | *Kolmogorov et al., 2019* | 2.6 | |
| Software, algorithm | Canu | *Koren et al., 2017* | 1.8 | |
| Software, algorithm | Miniasm | *Li, 2016* | 0.3 | |
| Software, algorithm | Guppy | Oxford Nanopore | 3.2.4 | |
| Software, algorithm | Medaka | Oxford Nanopore | 0.9.1 | |
| Software, algorithm | Minimap2 | *Li, 2016* | 2.17 | |
| Software, algorithm | SAMtools | *Li et al., 2009* | 1.12 | |
| Software, algorithm | Racon | *Vaser et al., 2017* | 1.4.3 | |
| Software, algorithm | BUSCO | *Simão et al., 2015* | 3.0.2 | |
| Software, algorithm | BUSCO | *Seppey et al., 2019* | 4.0.6 | |
| Software, algorithm | Purge_haplotigs | *Roach et al., 2018* | 1.1.1 | |
| Software, algorithm | npScarf | *Cao et al., 2017* | 1.9-2b | |
| Software, algorithm | Pilon | *Walker et al., 2014* | 1.23 | |
| Software, algorithm | BLAST | *Altschul et al., 1990* | 2.10.0 | |
| Software, algorithm | SPAdes | *Bankevich et al., 2012* | 3.11.1 | |
| Software, algorithm | FMLRC | *Wang et al., 2018* | 1.0.0 | |
| Software, algorithm | LINKS | *Warren et al., 2015* | 1.8.7 | |
| Software, algorithm | RepeatMasker | *Smit et al., 2013* | 4.1.0 | |
| Software, algorithm | Dfam repeat databse | *Hubley et al., 2016* | 3.1 | Library for RepeatMasker |
| Software, algorithm | RepBase RepeatMasker edition | *Bao et al., 2015* | 20181026 | Library for RepeatMasker |
| Software, algorithm | cross_match | *Green, 2009* | 1.090518 | |
| Software, algorithm | Tandem Repeat Finder | *Benson, 1999* | 4.0.9 | |
| Software, algorithm | Bioawk | *Li, 2017* | 1.0 | |
| Software, algorithm | GenomeScope | *Vurture et al., 2017* | 1.0.0 | |
| Software, algorithm | Jellyfish | *Marçais and Kingsford, 2011* | 2.2.3 | |
| Software, algorithm | Sambamba | *Tarasov et al., 2015* | 0.8.0 | |
| Software, algorithm | PEPPER-Margin-Deepvariant | *Shafin et al., 2021* | 0.4 | |
| Software, algorithm | BCFtools | *Li, 2011* | 1.12 | |
| Software, algorithm | Merqury | *Rhie et al., 2020* | 1.3 | |
| Software, algorithm | Pomoxis | Oxford Nanopore | 0.3.7 | |
| Software, algorithm | bedtools | *Quinlan and Hall, 2010* | 2.30.0 | |
| Software, algorithm | HALtools | *Hickey et al., 2013* | 2.1 | |
| Software, algorithm | Integrative Genomics Viewer | *Robinson et al., 2011b* | 2.9.4 | |
| Software, algorithm | MAFFT | *Katoh and Standley, 2013* | 7.453 | |
| Software, algorithm | RAxML-NG | *Kozlov et al., 2019* | 0.9.0 | |
| Software, algorithm | ASTRAL-MP | *Yin et al., 2019* | 5.14.7 | |
| Software, algorithm | ForceAtlas2 | *Jacomy et al., 2014* | | Implemented in R package https://github.com/analyxcompany/ForceAtlas2 |

*Continued on next page*

*Continued*

| Reagent type (species) or resource | Designation | Source or reference | Identifiers | Additional information |
|---|---|---|---|---|
| Software, algorithm | ape | *Paradis and Schliep, 2019* | 5.4.1 | R package |
| Software, algorithm | Docker | docker.com | | |
| Software, algorithm | Singularity | sylabs.io | | |

## Taxon sampling and sample collection

The selection of species used for this study was driven by several key objectives. First, we aimed to provide data for ongoing research projects. Second, we aimed to supplement existing genomic data, both as a benchmarking resource against well-studied references (e.g. *D. melanogaster*) and to provide a technological update to some older assemblies (*Roy et al., 2010*). Third, we aimed to increase the phylogenetic and ecological diversity of publically available *Drosophila* genome assemblies.

In most cases, genomic DNA was collected from lab-raised flies, which were either derived from lines maintained at public *Drosophila* stock centers and individual labs or, in a few cases, from F1 or F2 progeny of flies recently collected in the wild. We collected specimens from the wild with standard fruit or mushroom-baited traps, sweep netting, and aspiration. We established isofemale lines from individual females collected using these baits unless otherwise specified (*Supplementary file 1*). For species difficult to culture in the lab (all *Lordiphosa* spp. except *Lo. clarofinis, D. sproati, D. murphyi, Le. varia, S. graminum*), either wild-caught flies or flies from a transient lab culture were used. In accordance with domestic and international shipping laws, these flies were either fixed in ethanol before transport (*Lordiphosa* spp., *D. subobscura, D. obscura, C. costata, D. littoralis, D. tristis, D. ambigua*) or transported with permits (P526P-15–02964 to D. Matute, P526P-20–02787 and P526P-19–01521 to A. Kopp, and Hawaii State permit I1302 to D. Price).

Of 101 total assemblies, we include 13 genomes assembled with re-analyzed sequences from *Miller et al., 2018*; 60 genomes from stock center lines or established lab cultures; 22 genomes from lab-raised flies derived from recent wild collections; and six genomes from wild-caught flies. Of note, 6 *Zaprionus* lines used in this study (*Z. africanus, Z. indianus, Z. tsacasi, Z. nigranus, Z. taronus*) were assembled by *Comeault et al., 2020*, but updated higher contiguity assemblies are provided with this manuscript with the exception of *Z. indianus* line 16GNV01 (see 'Alternative hybrid assembly process' section below). Details on each sample including (if available) line designations and collection information, are provided in *Table 1* and *Supplementary file 6*.

## DNA extraction and nanopore sequencing

A high molecular weight (HMW) genomic DNA (gDNA) extraction and ONT library prep was performed for each sample, with slight variation in the protocol through time and to deal with differences in sample quality or preservation. Here, we briefly describe a recommended general protocol for HMW gDNA extraction and library prep from 15 to 30 flies. This protocol is sufficient to reproduce all results from this manuscript at the same or higher levels of data quality. Detailed step-by-step instructions are provided at Protocols.io (see **Data availability**). We note one exception made necessary by sample availability and shipping laws. *Scaptomyza graminum* gDNA was extracted by using the Qiagen Blood and Cell Culture DNA Mini Kit (Qiagen, Germantown, MD) from 30 unfrozen flies and prepared with the ONT LSK109 kit (Oxford Nanopore, Oxford, UK) without any modifications to the manufacturer's instructions.

Genomic DNA was prepared from about 30 flash frozen or ethanol fixed adult flies. For non-inbred samples, we tried to use 15 flies or less to minimize the genetic diversity of the sample. In the absence of amplification, about 1.5–3 µg of input DNA is needed to prepare 3–4 library loads with the ONT LSK109 kit. Sufficient input DNA is particularly important when selecting for longer reads. Ethanol preserved samples were soaked in a rehydration buffer (400 mM NaCl, 20 mM Tris-HCl pH 8.0, 30 mM EDTA) for 30 min at room temperature (~23℃), dabbed dry with a Kimwipe, then frozen for 1 hr at −80℃ before extraction. Frozen flies were ground in 1.5 mL of homogenization buffer (0.1M NaCl, 30 mM Tris HCl pH 8.0, 10 mM EDTA, 0.5% Triton X-100) with a 2 mL Kimble (DWK Life Sciences, Millville, NJ) Kontes Dounce homogenizer. The homogenate was centrifuged for 5 min at

2000 $\times g$, the supernatant discarded by decanting, and the pellet resuspended in 100 µL of fresh homogenization buffer. This mixture was then added to a tube with 380 µL extraction buffer (0.1M Tris-HCl pH 8.0, 0.1M NaCl, 20 mM EDTA) along with 10 µL of 20 mg/mL Proteinase K (Thermo Fisher Scientific, Waltham, MA), 10 µL SDS (10% w/v), and 2 µL of 10 mg/mL RNAse A (Millipore Sigma, Hayward, CA). This tube was incubated at 50°C for 4 hr, with mixing at 30–60 min intervals by gentle inversion.

High-molecular-weight gDNA was purified with a standard phenol-chloroform extraction. The lysate was extracted twice with an equal volume of 25:24:1 v/v phenol chloroform isoamyl alcohol (Thermo Fisher Scientific, Waltham, MA) in a 2 mL light phase lock gel tube (Quantabio, Beverly, MA). Next, the aqueous layer was decanted into a fresh 2 mL phase lock gel tube then extracted once with an equal volume of chloroform (Millipore Sigma, Hayward, CA). The use of the phase lock gel tube reduces DNA shearing at this stage by minimizing pipette handling. HMW DNA was precipitated by adding 0.1 vol of 3M sodium acetate and 2.0–2.4 volumes of cold absolute ethanol. Gentle mixing resulted in the precipitation of a white, stringy clump of DNA, which was then transferred to a DNA LoBind tube (Eppendorf, Hamburg, Germany) and washed twice with 70% ethanol. After washing, the DNA was pelleted by centrifugation and all excess liquid removed from the tube. The pellet was allowed to air dry until the moment it became translucent, resuspended in 65 µL of 1× Tris-EDTA buffer on a heat block at 50°C for 60 min, then incubated for at least 48 hr at 4°C. After 48 hr, the viscous DNA solution was mixed by gentle pipetting with a P1000 tip. This controlled shearing step encourages resuspension of HMW DNA and improves library prep yield. DNA was quantified with Qubit (Thermo Fisher Scientific, Waltham, MA) and Nanodrop (Thermo Fisher Scientific, Waltham, MA) absorption ratios were checked to ensure 260/280 was greater than 1.8 and 260/230 was greater than 2.0.

The sequencing library was prepared following the ONT Ligation Sequencing Kit (SQK-LSK109) protocol, with two important modifications. First, we started with approximately 3 µg of input DNA, three times the amount recommended by the manufacturer. Second, we utilized a form of size-selective polymer precipitation (*Paithankar and Prasad, 1991*) with the Circulomics Short Read Eliminator (SRE) buffer (Circulomics, Baltimore, MD) plus centrifugation to isolate DNA instead of magnetic beads. We found this to be necessary because magnetic beads irreversibly clumped with viscous HMW gDNA, decreasing library yield and limiting read lengths. The manner in which this was performed was specific to the cleanup step. After the end-prep/repair step (New England Biolabs, Ipswich, MA), the SRE buffer was used according to the manufacturer's instructions. After adapter ligation, DNA was pelleted by centrifuging the sample at 10,000$\times g$ for 30 min without the addition of any reagents, since DNA readily precipitated upon addition of the ligation buffer. Ethanol washes were avoided past this step since ethanol will denature motor proteins in the prepared library. Instead, the DNA pellet was washed with 100 µL SFB or LFB (interchangeably) from the ligation sequencing kit instead of 70% ethanol. If library yield was sufficient (>50 ng/µL), the Circulomics SRE buffer was used for a final round of size selection, replacing the ethanol wash with LFB/SFB as described above. Of note, a cheaper and open-source alternative made with polyethylene glycol MW 8000 (PEG 8000), although less effective at size selection, to the SRE buffer is described by *Tyson, 2020* (dx.doi.org/10.17504/protocols.io.7euhjew). A 1:1 dilution of the PEG 8000 solution described in that protocol can be substituted for SFB or LFB in the washing steps described above.

The typical yield of a library prepared in this manner is in the range of 1–1.5 µg. Approximately 350 ng of the prepared library was loaded for each sequencing run. To maintain flow cell throughput and read length, flow cells were flushed every 8–16 hr with the ONT Flow Cell Wash Kit (EXP-WSH003) and reloaded with a fresh library.

## Obtaining short read datasets for polishing

We performed 2×150 bp Illumina sequencing for most of the strains that did not have publicly available short read data available. Illumina libraries were prepared from the same gDNA extractions as the Nanopore library for most samples, with some exceptions as described in *Supplementary file 1*. The libraries were prepared in either of two manners. For the majority of samples, sequencing libraries were prepared with a modified version of the Nextera DNA Library Kit (Illumina, San Diego, CA) protocol (*Baym et al., 2015*) and sequencing was performed by Admera Health on NextSeq 4000 or HiSeq 4000 machines. Alternatively, Illumina libraries were prepared with the KAPA Hyper DNA kit (Roche, Basel, Switzerland) according to the manufacturer's protocol and sequenced at the UNC

sequencing core on a HiSeq 4000 machine. In either case, all samples on a lane were uniquely dual indexed. Illumina sequencing was not performed for *D. equinoxialis*, *D. funebris*, *D. subpulchrella*, *D. tropicalis*, *Le. varia*, *Z. lachaisei*, *Z. taronus*, and the unidentified São Tomé mushroom feeder due to material unavailability (line extinction/culling). Details for each sample, including accession numbers for any public data used in this work, are provided in *Supplementary file 1*.

### Choice of long read assembly program

Flye v2.6 (*Kolmogorov et al., 2019*) was used due to its quick CPU runtime, low memory requirements, excellent assembly contiguity, and its consistent performance on benchmarking datasets (*Wick and Holt, 2020*). We additionally validated the performance of Flye for *Drosophila* genomes using Nanopore data previously generated by *Miller et al., 2018* and 60× depth of new Nanopore sequencing of the Berkeley Drosophila Genome Project ISO-1 strain of *D. melanogaster*. We assembled genomes with Flye v2.6 and Canu v1.8 (*Koren et al., 2017*) to evaluate simple benchmarks of assembly contiguity and run time and to provide a comparison to the Miniasm (*Li, 2016*) assemblies from *Miller et al., 2018* Canu produced relatively contiguous assemblies, but a single assembly took several days on a 92-core cloud server and even longer when a large number of extra-long (>50kb) reads were present in the data. This was determined to be too costly when scaled to >100 species. In addition to a much shorter (8–12 hr wall-clock time) runtime, Flye also produced significantly more contiguous assemblies than those reported by Miller et al. (*Figure 1—figure supplement 2*). Note, several new long read assemblers have been released and these assembly programs have been significantly updated since this work was performed. Assembler performance should be evaluated with up-to-date versions in any future work.

### Assembly and long read polishing

After Nanopore sequencing was performed, raw Nanopore data were basecalled with Guppy v3.2.4, using the high-accuracy caller (option: -c dna_r0.4.1_450bps_hac.cfg). Raw Nanopore data previously generated by *Miller et al., 2018* were processed in the same manner.

Next, basecalled reads were assembled using Flye v2.6 with default settings. Genome size estimates (option: —genomeSize) were obtained through a web search or taken from a closely related species. If no such information was available, an initial estimate of 200 Mb was used. The specific genome size estimate provided to Flye (separate from the one estimated later from BUSCO coverage) is provided in *Supplementary file 2*.

After generating a draft assembly, we performed long read polishing using Medaka following the developer's instructions (https://nanoporetech.github.io/medaka/draft_origin.html). Reads were aligned to the draft genome with Minimap2 v2.17 (*Li, 2016*) and parsed with SAMtools v1.12 (*Danecek et al., 2021*; *Li et al., 2009*) before each round of polishing (option: -ax ont). The draft was polished with two rounds of Racon v1.4.3 (*Vaser et al., 2017*) (options: -m8 -x 6 g 8 w 500) and then a single round of Medaka v0.9.1.

### Haplotig identification and removal

Next, we assessed each Medaka-polished assembly for the presence of duplicated haplotypes (haplotigs) using BUSCO v3.0.2 (*Simão et al., 2015*; *Waterhouse et al., 2018*) along with the OrthoDB v9 dipteran gene set (*Zdobnov et al., 2017*). If the BUSCO duplication rate exceeded 1%, haplotig identification and removal was performed, but on the draft assembly produced by Flye rather than the polished assembly. Purge_haplotigs v1.1.1 (*Roach et al., 2018*) was run on these sequences following the guidelines provided by the developer (https://bitbucket.org/mroachawri/purge_haplotigs). Illumina reads were mapped to the draft assembly with Minimap2 (option: -ax sr) to obtain read depth information. The optional clipping step was performed to remove overlapping (duplicate) contig ends. Finally, remaining contigs were re-scaffolded with Nanopore reads using npScarf v1.9-2b (*Cao et al., 2017*), with support from at least four long reads required to link two contigs (option: —support=4). These sequences were polished with Racon and Medaka as described above.

### Final polishing and decontamination

The Medaka-polished assembly was further polished with Illumina data and any contigs identified as microbial sequences were removed. Illumina reads were mapped to the draft assembly with

Minimap2 (option: -ax sr) and the assembly polished with Pilon v1.23 (—`fix snps,indels`) (*Walker et al., 2014*). If a genome did not have an accompanying short read dataset but Illumina reads were available from a different strain of the same species (*Supplementary file 1*), Pilon was run without correcting SNVs (option: —`fix indels`). We found that allowing Pilon to fix gaps or local misassemblies in default mode introduced large spurious indels in regions where short reads map poorly such as tandem repeats. These variants were not supported by long reads or by comparison to a reference assembly. Thus, we chose to use Pilon to only fix base-level errors.

## Assembly decontamination

After Pilon polishing, assembly completeness was assessed again with BUSCO v3.0.2. We used BLAST v2.10.0 (*Altschul et al., 1990*) to remove any contigs not associated with at least one BUSCO that were also of bacterial, protozoan, or fungal origin. Finally, any sequences flagged by the NCBI Contamination Screen were excluded or trimmed.

A flow chart outline of the full genome assembly process described here is provided in *Figure 7*.

## Alternative hybrid assembly process

*Zaprionus indianus* line 16GNV01 had insufficient Nanopore data for a Flye assembly. For this line only and to consolidate all assemblies as a single resource, the same genome assembly from *Comeault et al., 2020* is both reported here and associated with the NCBI BioProject associated with this work. An alternative assembly strategy was taken for this line. Briefly, short-read sequence data was assembled first using SPAdes v3.11.1 (*Bankevich et al., 2012*) using default parameters. Nanopore reads were corrected with Illumina data using FMLRC v.1.0.0 (*Wang et al., 2018*) and subsequently used to scaffold the SPAdes assembly using LINKS v.1.8.7 (*Warren et al., 2015*) using the recommended iterative approach of 33 iterations with incrementally increasing *k*-mer distance threshold. The resulting scaffolds were polished with four rounds of Racon followed by four rounds of Pilon (but without Medaka) as described above.

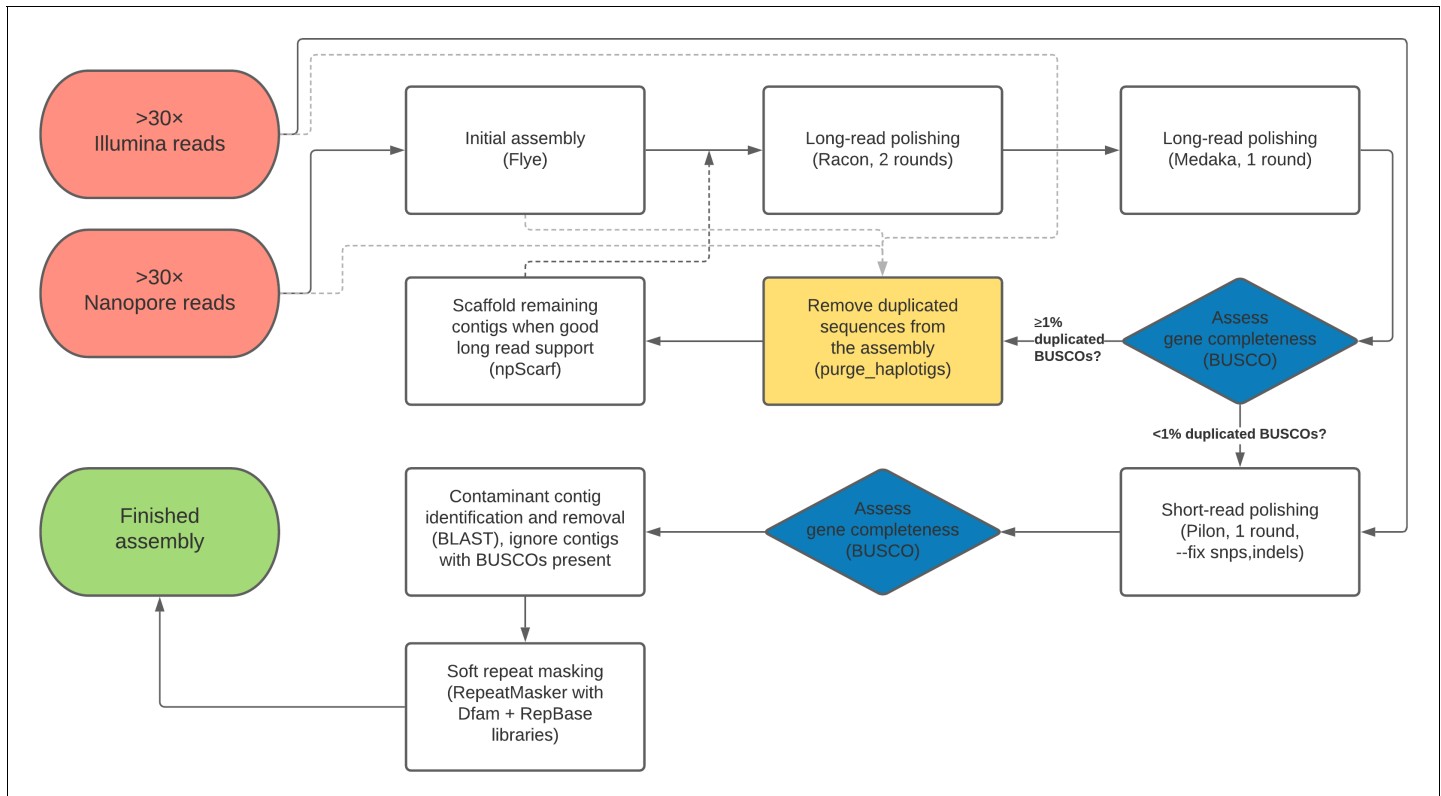

**Figure 7.** Flow chart depiction of the assembly pipeline.

## Repeat annotation and masking

Each draft assembly was soft repeat masked with RepeatMasker v4.1.0 (*Smit et al., 2013*) at medium sensitivity, with both Dfam 3.1 (*Hubley et al., 2016*) and RepBase RepeatMasker edition (*Bao et al., 2015*) repeat libraries installed (options: –species `Drosophila` –`xsmall`). Repeat-Masker was initialized with cross_match v1.090518 (*Green, 2009*) as the sequence search engine and Tandem Repeat Finder v4.0.9 (*Benson, 1999*).

## Genome size estimation

Genome size was estimated with Nanopore and Illumina data separately (*Supplementary file 2*). To estimate genome size from Illumina reads, we used the k-mer counting approach implemented in GenomeScope v1.0.0 (*Vurture et al., 2017*). Briefly, we followed the developer-provided workflow (https://github.com/schatzlab/genomescope) and generated a k-mer count histogram using a k-mer size of 21 (option: -m 21) with Jellyfish v2.2.3 (*Marçais and Kingsford, 2011*). The histogram was passed to the genomescope.R script to estimate the haploid genome size. We found these estimates to be somewhat unreliable, particularly when we tried to estimate genome size from a non-inbred sample. Due to this issue and because some samples were missing short read data, we took additional steps to estimate genome size from long reads.

Since the higher error rate of Nanopore reads (5–15%) precludes the use of k-mer based reference-free approaches for genome size estimation, we instead used regions annotated as a single-copy BUSCO gene to estimate genome size. Our rationale was that non-duplicated complete BUSCOs in each assembly could reasonably be assumed to be true single-copy markers and serve a similar function as unique k-mers for genome size estimation. Then, genome size can be roughly estimated from the depth of coverage across single-copy BUSCOs:

$$genome\ size\ =\ (total\ bases\ in\ Nanopore\ reads)/(coverage)$$

To perform this estimation, Nanopore reads were aligned to the coding sequences with Minimap2, only keeping primary alignments (options: –ax map-ont –`secondary=no`). Read depth was computed from genomic regions annotated as a single-copy BUSCO with SAMtools. If some proportion of the genome assembly was identified as non-fly and removed during the contaminant removal step, we adjusted the genome size estimate based on the total length of removed sequence:

$$genome\ size = (total\ bases\ in\ reads) * (1 - proportion\ of\ assembly\ removed)/(mean\ depth\ of\ coverage)$$

This assumes uniform Nanopore coverage across fly and contaminant sequence in the assembly and serves only as a rough approximation.

## Assessing assembly contiguity and completeness

Assembly contiguity statistics were computed using a series of custom shell and R scripts. Fasta files were parsed with Bioawk v1.0 (*Li, 2017*) and summary statistics were computed in the standard manner with the custom scripts. Contig N50 and NG50 were computed in the standard manner, in the latter case using the long-read based estimates of genome size. In addition to these statistics, we present contiguity in terms of *auN*. The *auN* statistic (*Li, 2020*) is the area under an *Nx* curve, and can be computed by multiplying the length of each contig ($L_i$) by the proportion of the assembled genome it accounts for ($L_i /\sum L_i$), then summing these values for all *i* contigs:

$$auN = \sum_i \left( L_i \frac{L_i}{\sum_j L_j} \right)$$

Contig N50 represents a single point on the *Nx* curve and may or may not be affected by assembly breaks, but *auN* is always sensitive to a break in the assembly. Therefore, *auN* is a fairer statistic for comparison between different versions of the same assembly.

Assembly completeness was assessed with BUSCO v4.0.6 (*Seppey et al., 2019*), using the OrthoDB v10 Diptera database (*Kriventseva et al., 2019*) (options: –m `geno` -l diptera_odb10 –augustus_species fly). Note, the BUSCO version used here is different from what was used during the assembly process. When this work was started, BUSCO v3 was the current version. Version 4 was released while the project was ongoing. For consistency, version three was used during the assembly

process for all assemblies, but the completeness of all final assemblies was assessed with BUSCO v4. For *D. equinoxialis* only, BUSCO v4.1.4 was used instead of v4.0.6 due to the presence of a bug that precluded the use of earlier versions.

## Computation of sample heterozygosity and sequence quality from long and short reads

Sample diversity was estimated by counting the number of non-reference single-nucleotide polymorphisms (SNPs) and indels, called separately from long and short reads. We mapped ONT reads to the finished genome with Minimap2 (option: -ax map-ont) then sorted the output with sambamba 0.8.0 (*Tarasov et al., 2015*). Variants were called with PEPPER-Margin-Deepvariant r0.4 (*Shafin et al., 2021*), following the developer's Singularity container-based 'Nanopore variant calling' instructions (https://github.com/kishwarshafin/pepper), to generate both variant call format (VCF) and banded genomic variant call format (gVCF) files, that is, a variant call file including intervals of invariant sites. Similarly, we mapped Illumina reads to each finished genome genome with Minimap2 (option: -ax sr), sorted and removed duplicates with sambamba, then performed variant calling, with output including all invariant sites, with BCFtools v1.12 (*Danecek et al., 2021*; *Li, 2011*). For both types of variant calls, we performed additional quality filtering using BCFtools. Only sites with minimum read depth 10, site quality score 30, and (if applicable) genotype quality score 30 filters were kept. The number of callable sites was estimated by adding the number of sites and the lengths of the intervals that passed these quality filters.

Sequence quality was estimated from variant calls following the standard workflow (e.g. *Koren et al., 2017*; *Solares et al., 2018*). Error was estimated by counting the number of non-reference variants (SNPs or indels), in either heterozygous or homozygous form, then dividing this count by the number of informative bases for variant calling: $P_{error}$ = (Number of variants)/(Number of callable sites). A Phred-scaled quality score (QV) was computed in the standard manner: $QV = -10 * \log_{10}(P_{error})$.

## Reference-free consensus quality scores

Reference-free quality score estimates were computed with Merqury v1.3 (*Rhie et al., 2020*), following the instructions provided by the developer on the GitHub repository (https://github.com/marbl/merqury). Briefly, we used the tools included with the installation of Merqury to estimate an optimal k-mer size for each genome assembly, at a collision rate of 0.001. Then, we built a k-mer database using the Illumina reads used to polish the genome assembly. Note, in some cases, Illumina reads from a different strain were used, and polishing was only used to correct indels. Finally, we ran the main Merqury script on the assembly of interest to estimate a genome-wide Phred-scaled consensus quality score (QV).

## Reference-based quality assessment

Reference-based quality score estimates were computed ONT Pomoxis v0.3.7 (https://github.com/nanoporetech/pomoxis) for assemblies where a well-annotated counterpart of not only the same species but the same strain was available through the NCBI RefSeq database. Gene and repeat annotations were downloaded from NCBI and coding regions, introns, intergenic regions, and repeats were parsed into BED formatted intervals with bedtools v2.30.0 (*Quinlan and Hall, 2010*). Introns were computed as the within-gene complement of exons, and intergenic regions were computed as the complement of genic regions. Then, we ran Pomoxis, which aligns each Nanopore-based assembly to the reference genome and assessed differences between the two genomes in 100 kb windows (option: -c 100000). Consensus quality was estimated by counting SNVs, insertions, and deletions and dividing the number of affected base pairs by the length of the alignment. This computation was performed separately for exons, introns, intergenic regions, repeats, and for the whole genome, using the genomic intervals described above.

For manual validation, we first used Pomoxis, as described above, to generate a list of all 1 bp or longer (option: -l 1) indel differences between our Nanopore-based assembly and the Release six assembly (*Hoskins et al., 2015*) of the *D. melanogaster* reference strain. The CA 8.2 MHAP version of the PacBio-based (*Kim et al., 2014*; *Koren et al., 2017*) *D. melanogaster* ISO-1 assembly was obtained from GenBank accession GCA_000778455.1. The iso1_onta2_quickmerge_scaffolds version

from *Solares et al., 2018* was downloaded from the GitHub repository associated with that project (https://github.com/danrdanny/Nanopore_ISO1). We aligned short reads, long reads, and each of the non-reference genomes to the Release six reference genome using Minimap2 (option, for short reads: -ax sr; for long reads: -ax map-ont; for genomes: -ax asm5), then sorted and parsed the output into BAM format with SAMtools. Repeat annotations for the Nanopore-based assembly were generated as described previously, then lifted over into reference coordinates. The liftover was performed with HALtools v2.1 (*Hickey et al., 2013*). Specifically, we aligned our Nanopore assembly to the Release six assembly with Minimap2 (options: –cx asm5 –cs long), converted the PAF alignment to MAF with Minimap2's paftools.js program, MAF alignment to HAL with HALtools' hal2maf program, and executed the liftover with HALtools' halLiftover program. Alignments and genomic intervals were viewed in the Integrative Genomics Viewer v2.9.4 (*Robinson et al., 2011b*).

## Species tree inference from BUSCO orthologs

We inferred species relationships using complete and single-copy orthologs identified by the BUSCO analysis. Amino acid sequences were used instead of nucleotide sequences to achieve better alignments in the face of high-sequence divergence (*Bininda-Emonds, 2005*). Out of 990 single-copy orthologs present in all assemblies, we randomly selected 250 to construct gene trees. The predicted protein sequence of each ortholog was aligned separately with MAFFT v7.453 (*Katoh and Standley, 2013*), using the E-INS-i algorithm (options: –ep 0 –genafpair –maxiterate 1000). Gene trees were inferred with RAxML-NG v0.9.0 (*Kozlov et al., 2019*), using the *Le and Gascuel, 2008* amino acid substitution model (options: –msa-format FASTA –data-type AA –model LG). The summary method ASTRAL-MP v.5.14.7 (*Yin et al., 2019*) was run with default settings to reconstruct the species tree. We note that this is not intended to be a definitive phylogenetic reconstruction of species relationships; see *Suvorov et al., 2021* for a time-calibrated phylogeny utilizing 158 drosophilid whole genomes.

## Analysis of chromosome organization

Syntenic comparisons were performed by representing the genome assemblies as paths through an undirected graph. The path each genome traverses can be considered a series of connections between single copy orthologous markers (i.e. BUSCOs). Using BUSCO v4 annotations for each final genome, we constructed a 3285 by 3285 symmetric adjacency matrix, with row and column headers (nodes) corresponding to 3285 possible BUSCOs from the diptera_odb10 database. Off-diagonal entries in each matrix (edges) were the number of times two single-copy BUSCOs were found as connected and immediate neighbors in the assemblies. Sequences of three or more BUSCOs were not considered. The graph was then visualized in two dimensions using the ForceAtlas2 graph layout algorithm (*Jacomy et al., 2014*) as implemented in the ForceAtlas2 R package (https://github.com/analyxcompany/ForceAtlas2). While this method is primarily designed for flexible, user-friendly tuning of graph visualization, it is similar in effect to other nonlinear dimensionality reduction techniques (*Böhm et al., 2020*). ForceAtlas2 was run with the settings: tolerance=1, gravity=1, iterations=3000. *D. equinoxialis* was omitted from this analysis due to the BUSCO v4 issues mentioned previously.

## Repeat content and genome size analysis

The contribution of repeat content to genome size variation in *Drosophila* was examined by comparing the number of bases in each genome annotated as a type of repeat (previously described) to the number of bases not annotated as repetitive sequence. Phylogenetic independent contrasts (*Felsenstein, 1985*) were computed for the counts of bases in both categories using the R package ape v5.4.1 (*Paradis and Schliep, 2019*) using the species tree described above with the root age set to 53 million years following the estimate in *Suvorov et al., 2021*.

## Compute containers

While the overall computational demands of this work were high, the unique computational challenge we faced was the variety of computational resources used for various stages of the assembly process. Assemblies took place across local servers, institutional clusters, and cloud computing resources. A key factor in ensuring reproducibility across computing environments was the use of computing containers, which is like a lightweight virtual machine that can be customized such that

sets of programs and their dependencies are packaged together. Specifically, we used the programs Docker and Singularity to manage containers. These programs allow containers to be built and packaged as an image file which is transferred to another computer. A Dockerfile, a text file containing instructions to set up an image, is used to select the Linux operating system and the suite of programs to be installed within a Docker container. Singularity is used to package the Docker container as an image file that can be transferred to and used in a cluster or cloud environment without the need for administrative permissions. Standard commands are then run inside the container environment. The files and instructions necessary to build these containers, which will allow for the exact reproduction of the computing environment in which this work was performed, are provided at: https://github.com/flyseq/drosophila_assembly_pipelines, (copy archived at swh:1:rev:4e40d28d0bdcd1bc7e4eabb7709f301df9ad7ead, *Kim, 2021*). We hope these files will facilitate the work of researchers new to Nanopore sequencing or the genome assembly process.

## Acknowledgements

We thank Brandon Cooper, Antonio Serrato-Capuchina, and David Turissini for help with collections and field logistics; Sarah CR Elgin, Wilson Leung, Elena Gracheva, and Sophia Bieser for help with modENCODE fly lines; Jonathan Chang for helpful discussions about phylogenetic methods; Charlotte Helfrich-Förster for providing lab resources for G Manoli and E Bertolini; and John Tyson along with the staff at Circulomics, in particular Kelvin Liu and Michelle Kim, for many illuminating discussions about long read library prep and sequencing.

## Additional information

### Funding

| Funder | Grant reference number | Author |
|---|---|---|
| National Institute of General Medical Sciences | F32GM135998 | Bernard Y Kim |
| National Institute of General Medical Sciences | R35GM118165 | Dmitri A Petrov |
| National Institute of Diabetes and Digestive and Kidney Diseases | K01DK119582 | Jeremy R Wang |
| National Science Foundation | DEB-1457707 | Corbin D Jones |
| National Institute of General Medical Sciences | R01GM121750 | Daniel R Matute |
| National Institute of General Medical Sciences | R01GM125715 | Daniel R Matute |
| Google | Google Cloud Research Credits | Bernard Y Kim Jeremy R Wang |
| National Institute of General Medical Sciences | R35GM122592 | Artyom Kopp |
| National Institute of General Medical Sciences | R35GM119816 | Noah Whiteman |
| Uehara Memorial Foundation | 201931028 | Teruyuki Matsunaga |
| Ministry of Education, Science and Technological Development of the Republic of Serbia | 451-03-68/2020-14/200178 | Marina Stamenković-Radak Mihailo Jelić Marija Savić Veselinović |
| Ministry of Education, Science and Technological Development of the Republic of Serbia | 451-03-68/2020-14/200007 | Marija Tanasković Pavle Erić |
| National Natural Science Foundation of China | 32060112 | Jian-Jun Gao |
| Japan Society for the Promotion of Science | JP18K06383 | Masayoshi Watada |

| Horizon 2020 - Research and Innovation Framework Programme | 765937-CINCHRON | Giulia Manoli Enrico Bertolini |
| --- | --- | --- |
| Czech Science Foundation | 19-13381S | Vladimír Košťál |
| Japan Society for the Promotion of Science | JP19H03276 | Aya Takahashi |
| National Science Foundation | 1345247 | Donald K Price |

The funders had no role in study design, data collection and interpretation, or the decision to submit the work for publication.

### Author contributions

Bernard Y Kim, Conceptualization, Resources, Data curation, Software, Formal analysis, Funding acquisition, Validation, Investigation, Visualization, Methodology, Writing - original draft, Project administration, Writing - review and editing; Jeremy R Wang, Conceptualization, Resources, Data curation, Software, Investigation, Methodology, Writing - original draft, Writing - review and editing; Danny E Miller, Conceptualization, Data curation, Investigation, Methodology, Writing - original draft, Writing - review and editing; Olga Barmina, Resources, Investigation; Emily Delaney, Aaron A Comeault, Resources, Data curation, Investigation, Writing - review and editing; Ammon Thompson, Data curation, Validation; David Peede, Emmanuel RR D'Agostino, Molly Zych, Investigation; Julianne Pelaez, Resources, Validation, Writing - review and editing; Jessica M Aguilar, Diler Haji, Teruyuki Matsunaga, Marina Stamenković-Radak, Mihailo Jelić, Marija Savić Veselinović, Marija Tanasković, Pavle Erić, Jian-Jun Gao, Masanori J Toda, Hideaki Watabe, Masayoshi Watada, Jeremy S Davis, Vladimír Košťál, Resources, Writing - review and editing; Ellie E Armstrong, Resources, Data curation, Methodology, Writing - review and editing; Yoshitaka Ogawa, Resources, Investigation, Methodology; Takehiro K Katoh, Giulia Manoli, Enrico Bertolini, Resources; Leonie C Moyle, R Scott Hawley, Resources, Supervision, Writing - review and editing; Aya Takahashi, Resources, Data curation, Supervision, Investigation, Writing - review and editing; Corbin D Jones, Resources, Supervision, Funding acquisition, Writing - review and editing; Donald K Price, Resources, Data curation, Funding acquisition, Writing - review and editing; Noah Whiteman, Conceptualization, Resources, Supervision, Funding acquisition, Project administration, Writing - review and editing; Artyom Kopp, Conceptualization, Resources, Supervision, Funding acquisition, Visualization, Project administration, Writing - review and editing; Daniel R Matute, Conceptualization, Resources, Supervision, Funding acquisition, Methodology, Writing - original draft, Project administration, Writing - review and editing; Dmitri A Petrov, Conceptualization, Resources, Formal analysis, Supervision, Funding acquisition, Visualization, Methodology, Writing - original draft, Project administration, Writing - review and editing

### Author ORCIDs

Bernard Y Kim ⓘ https://orcid.org/0000-0002-5025-1292
Jeremy R Wang ⓘ https://orcid.org/0000-0002-0673-9418
Emily Delaney ⓘ https://orcid.org/0000-0003-3609-5702
Aaron A Comeault ⓘ http://orcid.org/0000-0003-3954-2416
David Peede ⓘ http://orcid.org/0000-0002-4826-0464
Teruyuki Matsunaga ⓘ http://orcid.org/0000-0002-6433-622X
Mihailo Jelić ⓘ http://orcid.org/0000-0002-1637-0933
Pavle Erić ⓘ http://orcid.org/0000-0002-0053-1982
Masanori J Toda ⓘ http://orcid.org/0000-0003-0158-1858
Aya Takahashi ⓘ http://orcid.org/0000-0002-8391-7417
Noah Whiteman ⓘ https://orcid.org/0000-0003-1448-4678
Artyom Kopp ⓘ http://orcid.org/0000-0001-5224-0741
Dmitri A Petrov ⓘ https://orcid.org/0000-0002-3664-9130

### Decision letter and Author response

Decision letter https://doi.org/10.7554/eLife.66405.sa1

Author response https://doi.org/10.7554/eLife.66405.sa2

## Additional files

**Supplementary files**

• Supplementary file 1. Detailed information on both long-read and short-read data used for this project, including accession numbers if publicly available data were used for assembly.

• Supplementary file 2. Assembly summary statistics and genome size estimates.

• Supplementary file 3. Counts of SNPs, indels, and per-site heterozygosity estimated from both long reads and short reads.

• Supplementary file 4. Consensus quality scores estimated with reference-free and reference-based methods.

• Supplementary file 5. Characterization of all coding sequence indel differences between Nanopore and Release six reference *D. melanogaster* assemblies.

• Supplementary file 6. Detailed sample information.

• Transparent reporting form

### Data availability

All sequencing data and assemblies generated by this study are deposited at NCBI SRA and GenBank under NCBI BioProject PRJNA675888. Accession numbers for all data used but not generated by this study are provided in the supporting files. Dockerfiles and scripts for reproducing pipelines and analyses are provided on GitHub (https://github.com/flyseq/drosophila_assembly_pipelines; copy archived at https://archive.softwareheritage.org/swh:1:rev:4e40d28d0bdcd1bc7e4eabb7709f301df9ad7ead). A detailed wet lab protocol is provided at https://Protocols.io (https://doi.org/10.17504/protocols.io.bdfqi3mw).

The following dataset was generated:

| Author(s) | Year | Dataset title | Dataset URL | Database and Identifier |
|---|---|---|---|---|
| Kim BY, Wang JR, Kim BY, Wang JR | 2020 | Nanopore-based assembly of many drosophilid genomes | https://www.ncbi.nlm.nih.gov/bioproject/?term=prjna675888 | NCBI BioProject, PRJNA675888 |

The following previously published datasets were used:

| Author(s) | Year | Dataset title | Dataset URL | Database and Identifier |
|---|---|---|---|---|
| Miller DE | 2018 | Sequencing and assembly of 14 *Drosophila* species | https://www.ncbi.nlm.nih.gov/bioproject/PRJNA427774 | NCBI BioProject, PRJNA427774 |
| The Drosophila modENCODE Project | 2011 | modENCODE *Drosophila* reference genome sequencing (fruit flies) | https://www.ncbi.nlm.nih.gov/bioproject/63477 | NCBI BioProject, 63477 |
| Yang H | 2018 | DNA-seq of sexed *Drosophila* grimshawi, *Drosophila* silvestris, and *Drosophila* heteroneura | https://www.ncbi.nlm.nih.gov/bioproject/PRJNA484408 | NCBI BioProject, PRJNA484408 |
| Bronski M | 2019 | *Drosophila* montium Species Group Genomes Project | https://www.ncbi.nlm.nih.gov/bioproject/PRJNA554346 | NCBI BioProject, PRJNA554346 |
| Rane R | 2018 | Invertebrate sample from *Drosophila* repleta | https://www.ncbi.nlm.nih.gov/bioproject/476692 | NCBI BioProject, 476692 |
| Turissini D | 2017 | Fly lines | https://www.ncbi.nlm.nih.gov/bioproject/395473 | NCBI BioProject, 395473 |
| National Institute of | 2016 | Genome sequences of 10 | https://www.ncbi.nlm. | NCBI BioProject, |

| | | | | |
|---|---|---|---|---|
| Genetics [Japan] | | *Drosophila* species | nih.gov/bioproject/PRJDB4817 | PRJDB4817 |
| Ellison C | 2019 | Raw genomic sequencing data from 16 *Drosophila* species | https://www.ncbi.nlm.nih.gov/bioproject/PRJNA550077 | NCBI BioProject, PRJNA550077 |

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
