## [Decision Letter]

**Acceptance summary:**

*Drosophila* species have long served as an important model system for genetics and genomics. The authors have developed an important community resource of high standard genomes for many species across the *Drosophila* clade. This resource will serve to empower the next generation of *Drosophila* research and provides an important road map for similar efforts in other groups of organisms.

**Decision letter after peer review:**

Thank you for submitting your article "Highly contiguous assemblies of 101 drosophilid genomes" for consideration by *eLife*. Your article has been reviewed by 3 peer reviewers, and the evaluation has been overseen by a Reviewing Editor and Patricia Wittkopp as the Senior Editor. The following individual involved in review of your submission has agreed to reveal their identity: Timothy B Sackton (Reviewer #2).

Essential revisions:

The reviewers and we appreciated the impressive community resource that has been carefully brought together and that this acts as an important road map for other clade-wide sequencing efforts.

The strong points of consensus that emerge across the reviews, and in the discussion with the reviewers, were:

1) The need for base accuracy to be incorporated into the quality metrics discussed.

2) The need to place the work into a broad context of large-scale genome efforts to expand the readership outside of *Drosophila* researchers and show the expanding wave of community efforts focused on high-quality reference genomes. Suggestions included the vertebrate genome project and the earth biogenome project.

3) The reviewers all noted that acknowledgement of previous work in Drosophilidae was lacking. *Drosophila* researches have clearly been leading this charge for a while and it would be good to place the current effort into prior work more thoroughly.

4) While annotation of these genomes is beyond the scope of the current paper, it would be good to include more of a discussion of the planned road map for annotation going forward.

These points all should be addressed in a revised version. Below we have copied each review in its entirety. The reviewers have read each other's reviews and agree that all of the points are reasonable. Please provide a point by point response to the reviews, with a focus on the above broad goals.

*Reviewer #1 (Recommendations for the authors):*

– Data availability. I found all (or at least most) of the raw sequencing data. I couldn't find assembly accessions.

– Guidance for future sequencing work

A back of the envelope calculation based on typical core prices for PacBio Sequel II CLR sequencing and Illumina Novaseq PE150 reads suggest a price total price of $14.50 / GB (< $2,175 / 150GB) and $9 / GB (< $6,750 / 750 GB) respectively. To sample a 150MB genome at 100-fold depth (15GB) for both platforms would seem to imply a cost of ~$350 (comparable to the price cited by the authors) or less. Accepting lower depth would lead to concomittant decreases in price / genome. The average sequencing depth for this project appears from Table S1 to be 9.8GB for long reads and 12.8GB for short reads, implying that the PacBio approach might be a bit cheaper. I don't think a detailed price analysis is necessary (or even advisable), but communicating the fact that the authors' approach is one of at least two more or less equally viable approaches would be both valuable and accurate.

– Quality control metrics: sequencing error and sample polymorphism

A description of the consensus error rate of the assemblies would be an important piece of documentation serving two purposes. It permits users to quantify the amount of error they might expect from this particular resource. Relatedly, since many of these samples are conducted on strains for which near isogenic samples are difficult to acquire, measuring and reporting the heterozygosity would help guide users as to the extent of this property of the material from which the assembly was derived, especially if such users intend to use the reference strains described here for conducting genetic work.

– Context via comparison to existing resources

Existing resources for highly contiguous assemblies (operatively, contig N50 > 1MB)

– The total number of existing assemblies is already at least 75 with N50 >= 1MB (57 from NCBI[1], 13 from Miller[2], 4 from Comeault[3], 1 from Rezvykh[4]).

– Obviously, this includes a lot of within-species samples (especially for *D. melanogaster*, including many assemblies of the reference strain) and many assemblies that have been subsequently improved by the authors. However, the authors are also sampling the same species multiple times (including the *D. melanogaster* reference strain) to their total count, so this is at least consistent with their counting.

– Importantly, this resource triples unique species possessing highly contiguous assemblies from 34 to 102 and expands species group representation from 8 to 15. Although, the quinaria group, represented by D. innubila, already existed and wasn't sampled again here, they re-sample 7 of the previous 8 species groups and add 7 more.

– As far as I can tell, until now, the most distant relative of *D. melanogaster* within Drosophilidae with a highly contiguous genome was Scaptodrosophila lebanonensis, which is in the tribe Colocasiomyini. The manuscript adds two additional members of this tribe, Leucophenga varia and Chymomyza costata. So, the number of distant relatives is tripled, but the actual phylogenetic breadth isn't (if my understanding of Drosophilid taxonomy is correct, and it may not be)

Scholarship

The $1,000 Nanopore genome was cited by both [2] and [5].

"Future work to improve biological and taxonomic diversity, particularly for species difficult to culture, should employ single fly sequencing and assembly workflows (Adams et al., 2020)."

An earlier long read precedent can be found in [6].

In addition to enumerating other high-quality *Drosophila* genomes that already exist, it would be extremely useful to users to see a comparison of the quality of the resources when they have published descriptions (in order to guide authors as to what types of contiguity, completeness, and error they can expect, especially in comparison to this work). At the least, I think the authors should put their work in the context of best previous assemblies for each species (an operative definition of contig N50 > 1 Mb seems consistent with their own thinking), particularly when that work has been published. To be consistent with their own accounting, they might even consider addressing works like [7], and species that have experienced extensive high quality sequencing effort like *D. obscura*, *D. simulans*, and *D. pseudoobscura*. I have attached a table with entries corresponding to every species in Table S2 as well as additional species exceeding contig N50 of 1Mb, including citations and NCBI accession numbers when they could be found.

The authors have actually cited many of these assemblies in other work [8], including genomes that have yet to be published, so it would seem that the authors are aware of them and trust the quality enough to incorporate into their own work, so improving the scholarship should be straightforward.

Typo

– In the abstract. There are 93 species represented, not 95.

*Reviewer #2 (Recommendations for the authors):*

While this manuscript presents a large amount of valuable new data, and is inherently important for that reason alone, I believe that some key improvements and additional analyses could greatly strengthen this manuscript and really improve the value to the community.

1) Improvements to quality metrics.

This paper reports genomes that are at the low-cost end of the cost/quality tradeoff in genome assembly. This is a extremely valuable contribution, because many other large-scale projects in genomics (most notably, the Vertebrate Genome Project) have focused on the other end of this spectrum. Yet, for many researchers, a low-cost way to produce 10 genomes from related species may be higher value than a "complete" assembly from one species. However, it remains somewhat unclear exactly how good these genomes are, beyond the observation that the gene space is largely complete, and contig N50s are generally high.

Therefore, I think the biggest and most important improvement that would increase the reach and usefulness of this manuscript is improvements to quality metrics. A recent preprint from the Vertebrate Genome Project team (Rhie et al., 2020; https://www.biorxiv.org/content/10.1101/2020.05.22.110833v1.full) provides a number of potentially useful quality metrics that may be worth considering applying here, although of course not all will be relevant to this project, and I realize the computational burden of trying to do everything could be large. Nonetheless, I think it is crucial to be able to give some sense of consensus quality, as base-level errors in assemblies has negative effects on many downstream applications. Based on Koren et al., 2019, there appears to be a large drop in likelihood of disrupting a gene due to a indel error between QV30 and QV40, and Rhie et al., 2020 has a lot more detail on various aspects of consensus quality metrics. I realize that many existing tools, e.g. Merqury (also Rhie et al., 2020, in Genome Biology) make the assumption that Illumina data is available for the same individual as the genome assembly, which is not universally true here (even in approximation, e.g. treating a strain as an individual). Still, some attempt to tackle this problem seems necessary, even if it cannot be done perfectly.

2) Assembly content

Related to the first point, basic descriptions of genome size (e.g., estimated from k-mers) would help to contextualize the resource produced, as would a definition of "near chromosome level contiguity" and validation of which of the newly reported assemblies here reach that threshold (especially as contig N50s vary by several orders of magnitude). Again, I don't think the VGP definitions are the only possible way to approach this question, but there is value in having some systematic summary of overall contiguity.

This paper does not describe the extent to which heterogametic sex chromosomes (not expected in all species based on sampling) or mitochondrial genomes are recovered. Presumably at least the presence of mt scaffolds is picked up in the NCBI screens, and is the kind of information that would be relatively straightforward to add to a table.

3) *Drosophila* focus.

There is a tension throughout this manuscript between describing a basically *Drosophila*-specific resource, and describing a more generally applicable approach to low-cost, clade-wide assembly. I think that the latter is really necessary and important, since not every community or group has the resources (in money, samples, compute) or desire (for their scientific questions) to use a "VGP-style" approach (with long reads, short reads, HiC, and optical mapping to produce as close to error-free chromosomal assemblies as possible). But the value of this manuscript as a blueprint for low-cost community genomics is somewhat limited by the *Drosophila*-centric nature of the results.

The most obvious *Drosophila*-specific assumptions are the availability of a inbred strain, and a genome size of 100-250 Mb or so, with a few exceptions. Notably, the assemblies of the larger genomes (and the ones derived from wild-caught flies) tend to be worse, with lower contig N50s and auN metrics, more contigs, and more fragmented or missing BUSCOs.

Of course it would be well beyond the scope of this manuscript to attempt to validate any of these approaches in other clades, or provide a simple recipe for how to assembly any possible genome. Nonetheless, it would certainly be possible to broaden the discussion, and be clearer in the text when certain statements are *Drosophila* specific (e.g., the $350 in sequencing costs assumes a genome on the order of 100-200 Mb).

4) Limitations of the existing resource and future prospects for improvement:

The genomes presented here do not include annotations, or any other form of supplemental resource such as whole genome alignments (e.g. Armstrong et al., 2020), and don't use HiC or any kind of scaffolding to obtain true chromosomal scaffolds. I think these are understandable and defensible choices, given the computational and technical requirements to extend this work in those directions. However, it may be valuable to discuss more explicitly what this resource is and is not. At a minimum, doing so could prevent the corresponding authors from receiving many emails asking where the gene annotations for species X are once this work is published.

*Reviewer #3 (Recommendations for the authors):*

This is really an impressive resource and has the potential to be widely used both in terms of the data itself and also the methodology. I have several suggestions that may improve the manuscript.

There are several species (willistoni, paulistorum, etc.) that are sequenced more than once without any reference to why. It might be useful to describe that somewhere (Table 1?). As a resource, it would be simpler to use if it was clear when one isolate per species vs. multiple were appropriate to use for analyses.

Lines 59-61 – a bit more detail about modifications here since you are writing methods last.

Lines 107-111 – I'm concerned about the conclusions drawn about repeat content based upon the way the data was analyzed. A comprehensive analysis of repeat content is probably beyond the scope of this manuscript, but without de novo characterization of repeat sequences, I'm worried that satellites and TEs in more distantly related species may be missed if they are lineage restricted. I believe TRF would get at this to a point, but maybe not robustly. Other software like RepeatModeler might be better. However, my suggestion is not that these genomes are individually de novo annotated for repeats. It is just that the conclusions about relationships between contiguity or genome size and repeat content are presented with more caveats.

Figure 2 – This figure is a bit difficult to intuit, so this is another place where more detail in the main text would be useful.

Figure 3 – though the authors argue that this tree is not meant as a robust measure of phylogenetic relationships, it would be nice to put some support values on the tree.

Line 336 – I think the bioawk people would appreciate a citation (though all I can find on the internet is to cite the github page).

Line 340 – Please describe the auN statistic (what is L and what are we summing over?) in more detail. The Li github page describes nicely.

---

## [Author Response]

Essential revisions:The reviewers and we appreciated the impressive community resource that has been carefully brought together and that this acts as an important road map for other clade-wide sequencing efforts.The strong points of consensus that emerge across the reviews, and in the discussion with the reviewers, were:1) The need for base accuracy to be incorporated into the quality metrics discussed.

The revised manuscript now contains several new sections with evaluations of sample diversity and base quality. In the text, see the new “Estimates of sample diversity”, “Estimates of sequence quality”, and “Nanopore-based assemblies are highly accurate in coding regions” sections in the Results and Discussion and their accompanying sections in the Methods. The figures associated with these new sections are: Figure 2 along with Figure 2—figure supplement 1, Figure 3 along with Figure 3—figure supplements 1-2, and Supplementary Files 3-5. As with the rest of the manuscript, sample scripts and any raw data underlying the figures is provided in the GitHub repository. We will briefly summarize our new results below.

As the reviewers recognized, a big challenge in conducting quality evaluation properly was the variation in usable data for each assembly. As a brief summary, Illumina data were generated for most strains, downloaded from NCBI for other strains, and for a small number of strains we did not have Illumina reads (due to strain extinction/culling in 2020). A limited number of these genomes had reference assemblies of the same strain. This limited the usefulness of both reference-free and reference-based methods to specific assemblies, for example, we could not effectively utilize Merqury on genomes without short reads. Because of this, we chose to utilize several complementary strategies to evaluate both sample diversity and accuracy using long reads, short reads, and reference assemblies so that: (1) each genome had at least one associated measurement of sample diversity and quality and (2) a broad enough evaluation was performed with each method such that readers could infer a sense of quality even if some data were missing.

Briefly, we first performed variant calling using long reads and short reads separately, to both get a sense of the heterozygosity in each line and to get a count of non-reference variants to estimate the error rate. The number of non-reference variants divided by the number of callable sites was used as a proxy for the error rate similar to how error rates were estimated by Solares et al., (2018) and Koren et al., (2017). Reference-free estimates of sequence quality were computed with Merqury, using any short read data used for the assembly (when available). We find that overall, Nanopore-based assemblies do not seem to meet the often-cited QV40 “reference quality” threshold, but this is not unexpected as the effectiveness of short-read mapping and polishing will vary greatly across the genome.

We therefore followed up on genome-wide quality assessments with reference-based comparisons for the few genomes where we assembled the same line as the NCBI reference genomes. These comparisons allowed us to directly assess how our assemblies differed from the reference genomes in specific genomic regions (CDS, introns, repeats, etc.).

These reference-based comparisons showed our assemblies to indeed be highly accurate in coding regions, with a few important subtleties to note. At first glance, the mean consensus quality for coding sequences is concerningly low, ranging between QV20-QV30 even for *D. melanogaster*. However, when assessing quality in genomic windows or gene by gene, it quickly becomes clear that most coding sequences in the Nanopore assemblies are identical to the reference genomes.

After seeing this pattern, we thought to perform a manual assessment of every indel difference between our *D. melanogaster* genome and the reference, bringing in other long-read assemblies of the reference strain to help determine whether these large indels might be real or artifacts of our assembly. We found that nearly all (99.7%) protein-coding genes in our *D. melanogaster* genome were identical to the reference in coding sequences, and out of the total coding sequence differences between the two genomes, 99% of “erroneous” bases were explained by large indels that were also supported by the other long-read assemblies.

In the “Next Steps” section, we point out these issues and recommend that future work add lower-coverage high-fidelity sequencing to improve genome assemblies outside of complex coding sequences.

2) The need to place the work into a broad context of large-scale genome efforts to expand the readership outside of *Drosophila* researchers and show the expanding wave of community efforts focused on high-quality reference genomes. Suggestions included the vertebrate genome project and the earth biogenome project.

We have carefully re-examined our manuscript based on the reviewer comments and agree that too many relevant background details were omitted for the sake of brevity. Significant changes have been applied to the Introduction and Results and Discussion to address this shortcoming.

The Introduction is now started by citing the growing wave of large-scale genome assembly projects, to show the timeliness of the work. We also clarify why a cheap approach to high-quality genome assembly is valuable: time and cost are still big limitations to generating clade-scale genome datasets if one is not a part of a well-funded consortium. Ultra-long reads play a key role in constructing high-quality genomes at low cost. We hope this will provide readers with a better sense of the timeliness and significance of this work.

In addition, we have also revised the Next Steps section with better descriptions of forthcoming additions to this resource (annotations and whole-genome alignments). Many of the other large genome assembly projects are building (or have built) similar resources; we hope to both let the reader know these resources are forthcoming but also that we are utilizing modern tools to do so. Specifics are provided below (Point 4).

3) The reviewers all noted that acknowledgement of previous work in Drosophilidae was lacking. *Drosophila* researches have clearly been leading this charge for a while and it would be good to place the current effort into prior work more thoroughly.

This is an important point and we certainly did not mean to downplay the important role that *Drosophilia* labs have played in modern genomics. We have added both historical context and more background on the current state of *Drosophila* genomes to address this shortcoming. The revised Introduction emphasizes the important roles of *D. melanogaster*, the 12 *Drosophila* genomes, and the modENCODE project. The quantity and quality of drosophilid genome assemblies that were available prior to this work are more clearly discussed.

Our original plan for adding background on existing drosophilid genome assemblies was to include a new supplementary table listing the genome assemblies currently available on NCBI alongside some quality comparisons. However, a recently released (in the time since this manuscript was submitted) review of arthropod genome assemblies by Hotaling et al., (2021) provides exactly this: lists of genomes, accessions, the technology used to build the assembly, and contiguity/completeness statistics. Seventy-six drosophilid genomes for 75 species (two *D. pseudoobscura* subspecies) are listed in a file provided at the following link:

https://github.com/pbfrandsen/insect_genome_assemblies

Instead of essentially duplicating this table for this manuscript, we decided to point the reader to Hotaling et al. We compare the contiguity and completeness of our genomes to the other drosophilid assemblies and show that, while not fully chromosome-level, our genomes are comparable to many of the best reference genomes (Figure 1—figure supplement 1).

4) While annotation of these genomes is beyond the scope of the current paper, it would be good to include more of a discussion of the planned road map for annotation going forward.

Related to the last point from Point 2, we have revised the “Next Steps” section to clearly state what kinds of resources we plan to release in the near future.

Briefly, our plan follows the well-established workflows of the UCSC Comparative Genomics toolkits, similar to some recently published genome consortium studies (e.g. Genereux et al., 2020 and Feng et al., 2020). We have a whole-genome alignment with ~170 genomes, built with the Progressive Cactus aligner, in hand and plan to release it shortly. The alignment will greatly facilitate other kinds of analyses. Using the HALtools and PHAST software, we have prepared pipelines for generating PhastCons, PhyloP, and GERP tracks. We have also set up LiftOff (same species) and Comparative Annotation Toolkit (different species) based pipelines as a first pass at gene annotation. In the select cases where clades do not contain a well-annotated representative (e.g. *Zaprionus*), we plan to generate RNA-seq data for gene annotation.

While this annotation strategy was chosen because we do not have the centralization, personnel, or funding of the large genome consortia, the plan is to make these resources easily accessible for anyone wishing to utilize them or to perform their own iterative improvements such as annotation for specific taxa. Like this study, we will release these resources along with carefully documented workflows.

These points all should be addressed in a revised version. Below we have copied each review in its entirety. The reviewers have read each other's reviews and agree that all of the points are reasonable. Please provide a point by point response to the reviews, with a focus on the above broad goals.Reviewer #1 (Recommendations for the authors):– Data availability. I found all (or at least most) of the raw sequencing data. I couldn't find assembly accessions.

The assemblies should now be available on NCBI. The assemblies were submitted to NCBI before the initial draft of this paper was submitted. We apologize for the delay; it seems COVID has significantly delayed the genome submission process.

– Guidance for future sequencing workA back of the envelope calculation based on typical core prices for PacBio Sequel II CLR sequencing and Illumina Novaseq PE150 reads suggest a price total price of $14.50 / GB (< $2,175 / 150GB) and $9 / GB (< $6,750 / 750 GB) respectively. To sample a 150MB genome at 100-fold depth (15GB) for both platforms would seem to imply a cost of ~$350 (comparable to the price cited by the authors) or less. Accepting lower depth would lead to concomitant decreases in price / genome. The average sequencing depth for this project appears from Table S1 to be 9.8GB for long reads and 12.8GB for short reads, implying that the PacBio approach might be a bit cheaper. I don't think a detailed price analysis is necessary (or even advisable), but communicating the fact that the authors' approach is one of at least two more or less equally viable approaches would be both valuable and accurate.

This is a great point. Although this manuscript and the accompanying methods are very specific to the Nanopore and Illumina hybrid approach, there are certainly other equally viable approaches to be considered if one were planning a similar project today. Long-read sequencing with PromethION or PacBio CLR would almost certainly result in a lower $/GB cost with comparable read lengths. Assembly quality would also benefit greatly from lower coverage of higher-accuracy long reads like PacBio HiFi.

We are hesitant to make a specific recommendation given there have been significant (one might say game-changing) improvements to Nanopore protocols, computational tools, and new flow cell types in the last year, with more promised to come soon. The best practices also depend on sample type, quality, and other factors that can vary from experiment to experiment. There are a few new developments available today that are worth mentioning. It is not uncommon to see Nanopore runs with half the data contained in reads 100kb or longer, and new kits are reported to have substantially increased (reportedly doubled) the throughput of Nanopore flow cells. Even Nanopore-only assemblies are far more accurate than before simply due to improved basecalling methods. For example, in internal testing of Nanopore-only assemblies, we have seen genome-wide consensus accuracy increase by more than 2-fold simply by updating the basecaller. Still, homopolymer indels and regions of poor short read mapping remain problematic in the final assembly.

Because of this, we think it is important (if possible) to start looking past the Nanopore+Illumina approach and at alternative strategies. A Nanopore ultra-long read and lower-coverage high-accuracy long read (e.g. HiFi) approach will be far more effective at improving accuracy outside of coding regions, and seems to be gaining traction in the literature. Combining ultra-long and high-fidelity long reads could be a similarly affordable and more future-proof alternative to Nanopore+Illumina.

We have added this discussion to the “Next Steps” section and hope it is useful to readers of the manuscript that are planning their own assembly project today.

– Quality control metrics: sequencing error and sample polymorphismA description of the consensus error rate of the assemblies would be an important piece of documentation serving two purposes. It permits users to quantify the amount of error they might expect from this particular resource. Relatedly, since many of these samples are conducted on strains for which near isogenic samples are difficult to acquire, measuring and reporting the heterozygosity would help guide users as to the extent of this property of the material from which the assembly was derived, especially if such users intend to use the reference strains described here for conducting genetic work.

We have added measures of sample diversity and sequence quality to the manuscript; please see Points 1 and 2 above for specific details.

– Context via comparison to existing resourcesExisting resources for highly contiguous assemblies (operatively, contig N50 > 1MB)– The total number of existing assemblies is already at least 75 with N50 >= 1MB (57 from NCBI[1], 13 from Miller[2], 4 from Comeault[3], 1 from Rezvykh[4]).– Obviously, this includes a lot of within-species samples (especially for *D. melanogaster*, including many assemblies of the reference strain) and many assemblies that have been subsequently improved by the authors. However, the authors are also sampling the same species multiple times (including the *D. melanogaster* reference strain) to their total count, so this is at least consistent with their counting.– Importantly, this resource triples unique species possessing highly contiguous assemblies from 34 to 102 and expands species group representation from 8 to 15. Although, the quinaria group, represented by *D. innubila*, already existed and wasn't sampled again here, they re-sample 7 of the previous 8 species groups and add 7 more.– As far as I can tell, until now, the most distant relative of *D. melanogaster* within Drosophilidae with a highly contiguous genome was Scaptodrosophila lebanonensis, which is in the tribe Colocasiomyini. The manuscript adds two additional members of this tribe, Leucophenga varia and Chymomyza costata. So, the number of distant relatives is tripled, but the actual phylogenetic breadth isn't (if my understanding of Drosophilid taxonomy is correct, and it may not be).

As described in Point 3 above, we added comparisons to genome assemblies on NCBI, using the summaries provided in the very recent review of long-read insect genomes by Hotaling et al., 2021. As there are several genome versions for several species, of different or the same strain, and because we have re-analyzed or added to data previously used by Miller et al., 2018 and Comeault et al., 2020 (the earlier versions are also not on NCBI), we opted to perform genome comparisons using only the representative genome on NCBI. This also made it easier since we directly utilize Hotaling et al.’s table for any comparisons.

Citations to other studies generating high-quality drosophilid genomes, including the references provided by the reviewer, have been added to the Introduction. We also substantially revised the “Taxon Sampling” and the “Next Steps” sections to better reflect how our genomes fit into the phylogeny relative to the ones that exist: that is, that genomes we assembled were mostly from already well-studied groups and that subfamily Steganinae (along with several other clades) remains poorly sampled.

ScholarshipThe $1,000 Nanopore genome was cited by both [2] and [5]."Future work to improve biological and taxonomic diversity, particularly for species difficult to culture, should employ single fly sequencing and assembly workflows (Adams et al., 2020)."An earlier long read precedent can be found in [6].

References to these studies have been added in the appropriate locations.

In addition to enumerating other high-quality *Drosophila* genomes that already exist, it would be extremely useful to users to see a comparison of the quality of the resources when they have published descriptions (in order to guide authors as to what types of contiguity, completeness, and error they can expect, especially in comparison to this work). At the least, I think the authors should put their work in the context of best previous assemblies for each species (an operative definition of contig N50 > 1 Mb seems consistent with their own thinking), particularly when that work has been published. To be consistent with their own accounting, they might even consider addressing works like [7], and species that have experienced extensive high quality sequencing effort like *D. obscura*, *D. simulans*, and *D. pseudoobscura*. I have attached a table with entries corresponding to every species in Table S2 as well as additional species exceeding contig N50 of 1Mb, including citations and NCBI accession numbers when they could be found.The authors have actually cited many of these assemblies in other work [8], including genomes that have yet to be published, so it would seem that the authors are aware of them and trust the quality enough to incorporate into their own work, so improving the scholarship should be straightforward.

As discussed in the Essential revisions, we now compare the contiguity and completeness of Nanopore assemblies to representative genomes currently available on NCBI. Additionally, we conduct reference-based quality assessments for Nanopore genomes that have an NCBI RefSeq counterpart of the same strain. This should give the reader a better sense of how our Nanopore-based assemblies stack up against the highest-quality reference genomes. Importantly, we find coding sequences in Nanopore assemblies to be nearly identical to NCBI RefSeq genomes.

Typo– In the abstract. There are 93 species represented, not 95.

We initially counted genomes of subspecies (*D. malerkotliana malerkotliana* and *D. malerkotliana pallens*, *D. pseudoananassae pseudoananassae* and *D. pseudoananassae nigrens*) as separate species. This was incorrect and the number has been revised to 93.

Reviewer #2 (Recommendations for the authors):While this manuscript presents a large amount of valuable new data, and is inherently important for that reason alone, I believe that some key improvements and additional analyses could greatly strengthen this manuscript and really improve the value to the community.1) Improvements to quality metrics.This paper reports genomes that are at the low-cost end of the cost/quality tradeoff in genome assembly. This is an extremely valuable contribution, because many other large-scale projects in genomics (most notably, the Vertebrate Genome Project) have focused on the other end of this spectrum. Yet, for many researchers, a low-cost way to produce 10 genomes from related species may be higher value than a "complete" assembly from one species. However, it remains somewhat unclear exactly how good these genomes are, beyond the observation that the gene space is largely complete, and contig N50s are generally high.Therefore, I think the biggest and most important improvement that would increase the reach and usefulness of this manuscript is improvements to quality metrics. A recent preprint from the Vertebrate Genome Project team (Rhie et al., 2020; https://www.biorxiv.org/content/10.1101/2020.05.22.110833v1.full) provides a number of potentially useful quality metrics that may be worth considering applying here, although of course not all will be relevant to this project, and I realize the computational burden of trying to do everything could be large. Nonetheless, I think it is crucial to be able to give some sense of consensus quality, as base-level errors in assemblies has negative effects on many downstream applications. Based on Koren et al., 2019, there appears to be a large drop in likelihood of disrupting a gene due to a indel error between QV30 and QV40, and Rhie et al., 2020 has a lot more detail on various aspects of consensus quality metrics. I realize that many existing tools, e.g. Merqury (also Rhie et al., 2020, in Genome Biology) make the assumption that Illumina data is available for the same individual as the genome assembly, which is not universally true here (even in approximation, e.g. treating a strain as an individual). Still, some attempt to tackle this problem seems necessary, even if it cannot be done perfectly.

Briefly, we now include both reference-free and reference-based (if possible) quality assessments of our genomes in the revised text. We found the latter to be highly informative given the non-uniform distribution of errors in the *Drosophila* genomes. Although overall sequence quality often appears to be quite far from the QV30-40 (i.e. reference quality) level, we show that most coding sequences have reference-quality accuracy and most protein-coding genes are unaffected by indel errors. Moreover, large indels, many of which could be real, have a disproportionate effect on the reference-based error rate, leading to large underestimates of true sequence quality.

2) Assembly contentRelated to the first point, basic descriptions of genome size (e.g., estimated from k-mers) would help to contextualize the resource produced, as would a definition of "near chromosome level contiguity" and validation of which of the newly reported assemblies here reach that threshold (especially as contig N50s vary by several orders of magnitude). Again, I don't think the VGP definitions are the only possible way to approach this question, but there is value in having some systematic summary of overall contiguity.

We now provide genome size estimates and genome quality measures (NGx curves, NG50) in Figure 1—figure supplement 3 and Supplementary File 2. Similar to the quality assessment, we ran into issues trying to estimate genome size from k-mers. Not all samples had short reads plus samples with high diversity and/or low coverage had unusual k-mer count histograms leading to odd genome size estimates. Although we have long read data for every genome, the noisiness of Nanopore read (5-15% error rates) precludes k-mer based estimation of genome size.

To get around these issues, we estimated genome size by mapping Nanopore reads back to each assembly and looking at depth of coverage over single-copy BUSCO genes. Our reasoning here was that both noisy long reads and short reads should be mappable to the reference genome, and that single-copy BUSCO genes could safely be assumed to be reliably mappable single-copy regions in each genome. Assuming this, genome size can be roughly estimated from the depth of coverage and the summed length of all Nanopore reads.

These estimates of genome size turned out to be surprisingly consistent with the size of the assemblies. In most cases they were slightly larger than the assembly size. Of course there are caveats here, the most significant one being how we handled datasets with lots of apparent contamination (e.g., the assemblies from wild-collected Hawaiian *Drosophila*). Here we assumed uniform coverage over all contigs and adjusted the genome size estimate by the proportion of the assembly identified as contaminant sequence.

This paper does not describe the extent to which heterogametic sex chromosomes (not expected in all species based on sampling) or mitochondrial genomes are recovered. Presumably at least the presence of mt scaffolds is picked up in the NCBI screens, and is the kind of information that would be relatively straightforward to add to a table.

Admittedly we did not adopt a rigorous way to identify either sex chromosomes and mitochondrial genomes/contigs from the beginning, although we certainly should have the data to do so for most samples. Another complication is that high-coverage contigs were tagged for removal when running purge_haplotigs, so it is possible that many mitochondrial sequences were removed. This was roughly consistent with the NCBI contamination screen as mtDNA did not show up for many of the genomes. Unfortunately, we have not retained the NCBI contamination screen files, so the information on mtDNA contigs is not as straightforward to recover. We attempted to run the contamination screen through Docker (https://github.com/NCBI-Hackathons/ContamFilter) but there is an issue with the NCBI databases the Docker image points to and we could not get it to run.

Despite these shortsighted initial decisions, we expect this information to be eventually released. At least one person we know of is started working on identifying the sex chromosomes after we released the data. This is not a formal collaboration so we do not have a timeline for this work. We are also planning to run the VGP mitochondrial assembly pipeline (mitoVGP) with our sequences and attempt to properly assemble the mitochondrial genomes. This is not ready to go into this manuscript, but we plan to include it in a future update.

3) *Drosophila* focus.There is a tension throughout this manuscript between describing a basically *Drosophila*-specific resource, and describing a more generally applicable approach to low-cost, clade-wide assembly. I think that the latter is really necessary and important, since not every community or group has the resources (in money, samples, compute) or desire (for their scientific questions) to use a "VGP-style" approach (with long reads, short reads, HiC, and optical mapping to produce as close to error-free chromosomal assemblies as possible). But, the value of this manuscript as a blueprint for low-cost community genomics is somewhat limited by the Drosophila-centric nature of the results.The most obvious *Drosophila*-specific assumptions are the availability of a inbred strain, and a genome size of 100-250 Mb or so, with a few exceptions. Notably, the assemblies of the larger genomes (and the ones derived from wild-caught flies) tend to be worse, with lower contig N50s and auN metrics, more contigs, and more fragmented or missing BUSCOs.Of course it would be well beyond the scope of this manuscript to attempt to validate any of these approaches in other clades, or provide a simple recipe for how to assembly any possible genome. Nonetheless, it would certainly be possible to broaden the discussion, and be clearer in the text when certain statements are *Drosophila* specific (e.g., the $350 in sequencing costs assumes a genome on the order of 100-200 Mb).

In the revised manuscript, we have tried to make it clearer that the $350 assembly is assuming *Drosophila*-specific parameters, both in terms of genome size and the amount of genomic DNA one might start the library prep process with. We have also improved the “Next Steps” to improve the discussion around sequencing species that cannot easily be reared in the lab or are only available as a few ethanol-preserved specimens. While the text remains somewhat specific to drosophilids, the discussion of the challenging samples we face and plans to address these challenges should have broader relevance to those not sequencing drosophilids.

4) Limitations of the existing resource and future prospects for improvement:The genomes presented here do not include annotations, or any other form of supplemental resource such as whole genome alignments (e.g. Armstrong et al., 2020), and don't use HiC or any kind of scaffolding to obtain true chromosomal scaffolds. I think these are understandable and defensible choices, given the computational and technical requirements to extend this work in those directions. However, it may be valuable to discuss more explicitly what this resource is and is not. At a minimum, doing so could prevent the corresponding authors from receiving many emails asking where the gene annotations for species X are once this work is published.

This is mostly addressed in our reply to Point 4, but to add a few specific comments here: we thought this was a very important point and have addressed it in the updated manuscript. We revised the “Next Steps” section to state clearly that whole genome alignments are forthcoming, and that whole genome alignments will allow us to start the annotation process in a logistically efficient manner.

Reviewer #3 (Recommendations for the authors):This is really an impressive resource and has the potential to be widely used both in terms of the data itself and also the methodology. I have several suggestions that may improve the manuscript.There are several species (willistoni, paulistorum, etc.) that are sequenced more than once without any reference to why. It might be useful to describe that somewhere (Table 1?). As a resource, it would be simpler to use if it was clear when one isolate per species vs. multiple were appropriate to use for analyses.

To be completely forthcoming, we ended up performing multiple assemblies for a few species without specifically intending to do so. Two *D. immigrans* strains were sequenced separately due to a miscommunication; *Z. tsacasi* car7-4 was originally misidentified as *Z. tuberculatus*; and the additional *D. willistoni* line (14030-0811.17) was originally sequenced for a grad student project in planned in South America, but this plan was disrupted by the pandemic. Separate lines of *D. teissieri, D. paulistorum,* and *Z. indianus* were sequenced because the lines appear to be reproductively isolated (observed by D. Matute).

We have tried to address any confusion the reader might experience by denoting a preferred assembly with an asterisk in Table 1 of the revised manuscript. Clarifying text regarding preferred assemblies has also been added to “Taxon Sampling.” Preferred assemblies were selected on the basis of contiguity, completeness, and whether Illumina data from the same sample was used for the assembly.

Lines 59-61 – a bit more detail about modifications here since you are writing methods last.

Added text clarifies the protocol modifications that improve the yield of ultra-long reads while improving library prep and sequencing efficiency (hence reducing costs).

Lines 107-111 – I'm concerned about the conclusions drawn about repeat content based upon the way the data was analyzed. A comprehensive analysis of repeat content is probably beyond the scope of this manuscript, but without de novo characterization of repeat sequences, I'm worried that satellites and TEs in more distantly related species may be missed if they are lineage restricted. I believe TRF would get at this to a point, but maybe not robustly. Other software like RepeatModeler might be better. However, my suggestion is not that these genomes are individually de novo annotated for repeats. It is just that the conclusions about relationships between contiguity or genome size and repeat content are presented with more caveats.

This is a good point. As mentioned, we would like other groups with more expertise in analyzing repeat content to be able to freely work on these data as that is not the focus of this work. We have added text to the “Repeat content” section to indicate that these results are conditional on what is currently in the repeat databases and that this resource is intended to be a starting point to improve repeat databases rather than a comprehensive characterization of repeats in drosophilid genomes.

Figure 2 – This figure is a bit difficult to intuit, so this is another place where more detail in the main text would be useful.

The main text has been revised to specifically state that the graph layout method is being used to create a map where BUSCOs that are connected in the assemblies will cluster together.

Figure 3 – though the authors argue that this tree is not meant as a robust measure of phylogenetic relationships, it would be nice to put some support values on the tree.

We have added local posterior probabilities (as reported by ASTRAL) to provide a measure of support for each node.

Line 336 – I think the bioawk people would appreciate a citation (though all I can find on the internet is to cite the github page).

This is our mistake, a citation to this tool should have been included. In the revised version of the manuscript, a citation to the GitHub page is included.

Line 340 – Please describe the auN statistic (what is L and what are we summing over?) in more detail. The Li github page describes nicely.

Based on Reviewer 2’s comments, we decided to de-emphasize *auN* as the main summary statistic for contiguity as most readers will have an immediate intuitive understanding of N50/NG50 but not *auN*. N50 and NG50 are now presented in Figure 1 and Figure 1—figure supplement 3, respectively.

Nevertheless, we still think *auN* provides the best comparison of contiguity between assemblies and plan to utilize it in future work, for instance, we plan to update these genomes with improved Nanopore basecalling algorithms. We have thus clarified the calculation of this statistic further in the Methods and still provide auN in Supplementary File 2.